# Boundary vector cells in the goldfish central telencephalon encode spatial information

**Lear Cohen**[1,2], **Ehud Vinepinsky**[3], **Opher Donchin**[1,2], **Ronen Segev** [1,2,4] *

**1** Department of Biomedical Engineering, Ben-Gurion University of the Negev, Beer-Sheva, Israel,
**2** Zlotowski Center for Neuroscience, Ben-Gurion University of the Negev, Beer-Sheva, Israel, **3** Institut de Biologie de l'École Normale Supérieure, Paris, France, **4** Department of Life Sciences, Ben-Gurion University of the Negev, Beer-Sheva, Israel

* ronensgv@bgu.ac.il

## Abstract

Navigation is one of the most fundamental cognitive skills for the survival of fish, the largest vertebrate class, and almost all other animal classes. Space encoding in single neurons is a critical component of the neural basis of navigation. To study this fundamental cognitive component in fish, we recorded the activity of neurons in the central area of the goldfish telencephalon while the fish were freely navigating in a quasi-2D water tank embedded in a 3D environment. We found spatially modulated neurons with firing patterns that gradually decreased with the distance of the fish from a boundary in each cell's preferred direction, resembling the boundary vector cells found in the mammalian subiculum. Many of these cells exhibited beta rhythm oscillations. This type of spatial representation in fish brains is unique among space-encoding cells in vertebrates and provides insights into spatial cognition in this lineage.

## Introduction

For most animals, the ability to locate themselves in the environment is crucial for survival [1–4]. This ability requires encoding information about self-position and locomotion [5,6]. Studies in navigating mammals have identified several types of neurons that encode self-position and locomotion in the hippocampal formation [5–9].

The place cell found in the hippocampus of rats [10] was the first spatial cell type to be identified. These cells are activated once the animal is in the cell's preferred location (place field) in the environment. After the discovery of place cells, the existence of boundary vector cells was hypothesized to explain the dependency of place cells on environment borders [11]. Later, boundary-vector cells that encode proximity to allocentric boundaries [12] were found, together with other spatial cell types in the brain areas connected to the hippocampus (e.g., the entorhinal cortex and the subiculum). These spatial cells include the grid cells [13], and the head direction cells [13], which encode the animal's allocentric head direction.

A recent study described the existence of single cells that encode spatial and kinematic features of the environment in the goldfish lateral pallium [14]. The homology of this brain region between fish and the mammalian hippocampal formation is supported by anatomical and lesion studies [15]. The findings in goldfish include cells that encode the environment's edges,

**Data Availability Statement:** All data in now available on GitHub: https://github.com/learco1510/BVC_Goldfish.git, and in Zenodo: https://zenodo.org/badge/latestdoi/575328476 DOI: 10.5281/zenodo.7684281.

**Funding:** We gratefully acknowledge financial support from THE ISRAEL SCIENCE FOUNDATION—FIRST Program (Grant no. 281/15 to RS and OD), THE ISRAEL SCIENCE FOUNDATION—FIRST Program (Grant no. 555/19 to RS), THE ISRAEL SCIENCE FOUNDATION (Grant no. 211/15 to RS), The Human Frontiers Science Foundation Grant (RGP0016/2019 to RS), and the Helmsley Charitable Trust through the Agricultural, Biological and Cognitive Robotics Initiative of Ben-Gurion University of the Negev (OD and RS). The funders had no role in study design, data collection and analysis, decision to publish, or preparation of the manuscript.

**Competing interests:** The authors have declared that no competing interests exist.

**Abbreviations:** Dc, large-celled subdivision of Dm; Dd, dorsal division of area dorsalis; Dld, dorsal subdivision of lateral division of area dorsalis; Dlv, ventral subdivision of lateral division of area dorsalis; Dlv-d, dorsal part of Dlv; Dlv-v, ventral part of Dlv; Dm, medial subdivision of area dorsalis; Dmc, caudal part of medial subdivision of area dorsalis; Dmr, rostral part of medial subdivision of area dorsalis; Vd, dorsal nucleus of area ventralis; Vdi, intermediate subnucleus of Vd; Vs, supracommissural nucleus of area ventralis; Vv, ventral nucleus of area ventralis.

the fish's head direction, speed, and velocity [14]. However, no evidence of neuronal activity in a specific place field was found, although place cells have been identified in many studies in mammals [16] and recently were also reported to exist in birds [17,18]. In addition, no evidence of rhythmic neural oscillations associated with navigation has been found in the goldfish comparable to those in the theta frequency range of the mammalian hippocampal formation [19]. Therefore, a comparative approach is needed to better understand the fundamental mechanisms of spatial cognition across vertebrates. This can shed light on whether the mammalian model is valid for all vertebrates or whether different classes evolved different computational schemes. For this purpose, it is crucial to determine how position is represented in the brain of fish, the largest vertebrate class.

One fundamental difference between fish and other vertebrates is the environment in which they navigate in. Navigation in aquatic environments is unique for several reasons. Fish navigate in full 3D, unlike terrestrial animals that are limited by the vertical dimension of their world. Furthermore, unlike flying animals, fish are subjected to a steep pressure gradient while navigating in the vertical dimension of their environment. This might imply that information about position in space is encoded in the fish brain differently than the locally activated cells shown to exist in 3D environments in bats [20] and rats [21].

To probe navigation in fish, we recorded multiple single-cell activities from the central area of the goldfish telencephalon. During the recordings, the goldfish could freely explore a water tank along its vertical and horizontal axes (see Fig 1A). The recorded site in the goldish brain is located in the central area of their telencephalon, although its boundaries and function remain somewhat undetermined [22–24]. Studies have, however, suggested that this area integrates information from multiple neighboring pallial regions [23]. We hypothesized that it would integrate representations of space from the entire telencephalon.

## Results

To better understand how space is represented in the teleost brain, we recorded the activity of multiple single cells in the goldfish telencephalon. Before the recordings, we trained the fish to swim continuously in a rectangular water tank (Fig 1A) measuring 0.7 m on the horizontal axis and 0.7 m on the vertical axis (0.2 m in the third, foreshortened axis) by feeding the fish in various places in the tank. In most cases after 1 to 2 weeks of training sessions (every other day), the fish became familiar with the water tank and adapted to exploring its entire environment freely. At the end of the last training session, we implanted an extracellular tetrodes and wireless recording system (see Materials and methods) in its central telencephalon (Fig 1B; additional histology examples are presented in S1 Fig). After a day of recovery from the implant surgery, we let the goldfish explore the quasi-2D water tank embedded in a 3D environment while recording neuronal activity and tracking its position. Neuronal activity was recorded using 3 to 4 tetrodes, and single cells were identified using spike sorting (Fig 1C–1E), ranging from 1 to 12 single cells per recording session and from 1 to 8 single cells per tetrode. Subsequently, we analyzed the associations between cell activity and the fish's trajectory (see Materials and methods).

The results showed that a considerable portion of the recorded cells had neural activity that was modulated by the fish's position in the environment (39 out of 123; i.e., about 32% of all the well-isolated units recorded in the 15 fish used in this study. See examples in Fig 2). Of the 39 spatially modulated cells, 4 cells had a diffuse firing pattern and were removed from further analyses. The remaining 35 spatially modulated cells were recorded from the brains of 7 of the 15 fish used in this study, ranging from 1 to 11 spatially modulated cells per fish during up to 5 recording sessions, with a maximum of 5 spatially modulated cells in a single experiment.

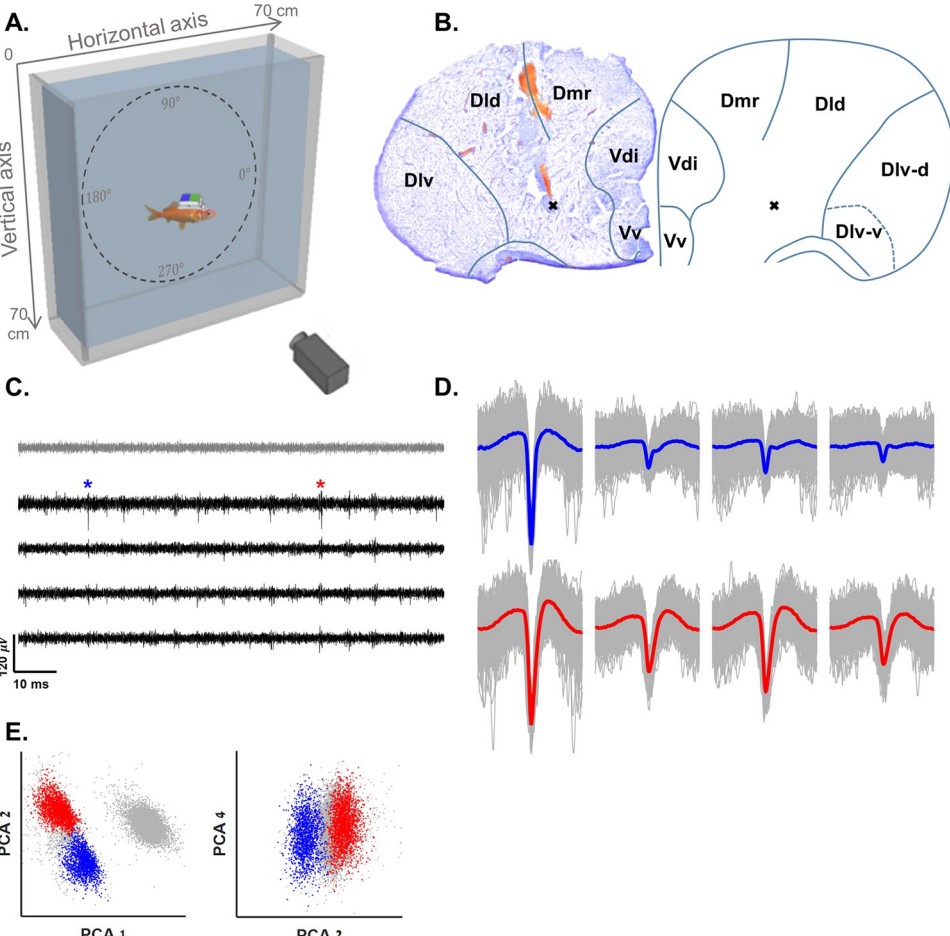

**Fig 1. Experimental setup and spike sorting. (A)** Schematic overview of the experimental setup: A fish swims freely in a water tank with a wireless recording system (see Materials and methods) mounted on its head. The fish's movements are recorded by a Raspberry Pi camera positioned in front of the tank. Dashed circle presents the directions used in all data analyses in this paper. We used camera-east as zero with an anti-clockwise progression. **(B)** Example of the recording site in the goldfish central telencephalon and the corresponding brain region (right panel, anatomical diagram based on [40]). Black x's show a location in which boundary vector cells were recorded. **(C)** Example of a raw recording from a tetrode (black traces) and a reference electrode (gray) in the fish's central telencephalon. Neural activity can be seen in the tetrode alone. Blue and red asterisks correspond to the blue and red clusters in panels D and E. The blue cluster corresponds to the cell presented in S2B Fig. The red cluster corresponds to the cell in Fig 2A–2D. **(D)** Waveforms of the two neurons after spike sorting. **(E)** Projections on the main principal components of the data from the tetrode of all spike candidates that crossed the threshold. Other clusters were not distinguishable from other multiunit activity and neural noise. The underlying data supporting panels D and E can be found in a file named **Fig 1_data.mat** (see Data Availability). Dld, dorsal subdivision of lateral division of area dorsalis; Dlv, ventral subdivision of lateral division of area dorsalis; Dlv-d, dorsal part of Dlv; Dlv-v, ventral part of Dlv; Dmr, rostral part of medial subdivision of area dorsalis; Vd, dorsal nucleus of area ventralis; Vdi, intermediate subnucleus of Vd; Vv, ventral nucleus of area ventralis.

This population of space encoding cells reported here can best be described as boundary vector cells. This encoding scheme is characterized by a gradually decreasing firing rate with the animal's distance from a boundary located at a specified direction. The presented cells, recorded from the central telencephalon of goldfish, had a gradual activity in a specified direction ($\Theta_{max}$; see Materials and methods) rather than activity that is localized in space (as in place cells) or tightly tuned to physical barriers (as in border cells). This makes them appear to resemble the mammalian boundary vector cells [12].

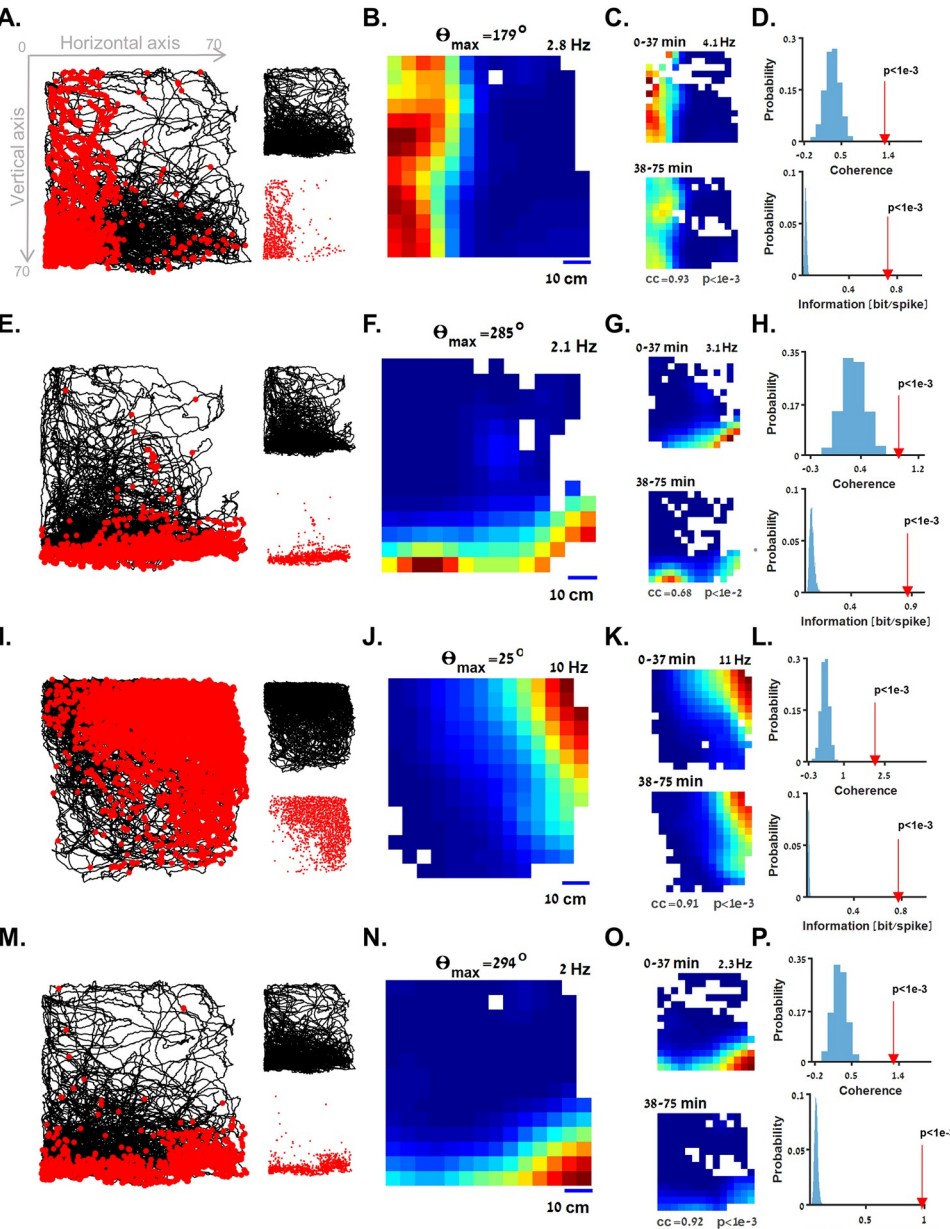

**Fig 2. Boundary vector cells in the goldfish brain. (A-D)** An example of a boundary vector cell tuned to distance from the left wall of the water tank (experimental setup is shown in Fig 1A). (**A**) Left: Fish trajectory (black curve) is presented together with the location of the fish when each spike of a single cell occurred (red dots). The neuron was mainly active when the fish was near the left wall of the tank. Right: trajectory (top panel) and spike locations (bottom panel). (**B**) Firing rate heatmap of the cell in A, color coded from dark blue (zero firing rate) to dark red (maximal firing rate, indicated on the upper right side of the panel). The preferred boundary direction of this cell is indicated ($\Theta_{max}$; see Materials and methods). The heatmap is occupancy corrected (see Materials and methods) to obtain a reliable estimate of the firing rate as a function of position. (**C**) In-session stability of the cell in A. The similar rate of the cell in the first (top panel) and second (bottom panel) halves of the experiment suggest stable activity within the recording session (correlation coefficient and *p*-value are indicated; see Materials and methods). (**D**) Spatial coherence (red arrow, top panel) and spatial information (red arrow, bottom panel) values of the cell in A are higher than those of 5,000 shuffled spike trains obtained from the same dataset (blue histograms; see Materials and methods). (**E-H**) Example of a boundary vector cell tuned to the bottom of the water tank. (**I-P**) Two other examples of boundary vector cells with a preferred boundary direction, which is neither horizontal nor vertical. The underlying data supporting all panels in this figure can be found in a file named **Fig2_data.mat** (see Data Availability).

Fig 2A–2D and S1 Video show an example of this type of cell whose firing rate gradually decreased with the distance of the fish from the left wall of the tank ($\Theta_{max} = 179^o$). This was manifested in its spiking activity (red dots, Fig 2A) throughout the fish's trajectory (black curve) as well as in the occupancy-corrected rate map in the water tank (Fig 2B).

To test statistically whether this neuron encoded components of position, we tested for in-session stability, spatial coherence, and for spatial information, as well as for crossing a spatial information threshold of 0.1 bits/spike (see Materials and methods). The similarity between the firing rate maps for the first (0 to 37 min) and the second (38 to 75 min) halves of the session indicated in-session stability (Fig 2C, correlation coefficient = 0.93, $p < 1 \times 10^{-3}$; see Materials and methods). The cell's spatial coherence (Fig 2D, top panel, red arrow) and spatial information (Fig 2D, bottom panel, red arrow) were higher than the corresponding values for 5,000 shuffled spike trains generated using an interspike interval (ISI) shuffling procedure (blue histograms; see Materials and methods).

Another example of a boundary vector cell tuned to distance from the bottom of the water tank ($\Theta_{max} = 285^o$) is presented in Fig 2E–2H. Additional examples of boundary vector cells with a horizontal (i.e., $\Theta_{max} = 15^o$ or $180^o \pm 15^o$) or a vertical (i.e., $\Theta_{max} = 90^o \pm 15^o$ or $270^o \pm 15^o$) preferred boundary direction are presented in S2B–S2N Fig, as well as in S2 and S3 Videos.

Not all boundary vector cells were tuned to boundaries in the vertical or horizontal directions. Rather, some cells had a firing pattern that gradually decreased with the distance of the fish from the corners of the water tank. An example of this type of cell is presented in Fig 2I–2L. For this cell, firing gradually decreased with distance from the top right corner ($\Theta_{max} = 25^o$) of the water tank. Other examples of boundary vector cells with a preferred boundary in the direction of the corners of the water tank are presented in Figs 2M–2P and S2O–S2W.

To assess the preferred boundary direction of the boundary vector cells, we calculated the 2D correlation coefficient of the cells' activity and the fish's position (Fig 3A, red dot, which corresponds to the cell presented in Fig 2A–2D). The correlation coefficient was then compared to the 2D correlation coefficients of 5,000 shuffled spike trains (grey dots, the 97.5 percentile is depicted; see Materials and methods). This analysis showed that the example cell (Fig 2A) had a left boundary tuning ($\Theta_{max} = 179^o$, the preferred and null directions and corresponding correlation coefficients are indicated; see Materials and methods). This was further confirmed by the spiking activity of the cell (red dots, Fig 3B) over the distance of the fish from the preferred boundary. This cell's tuning curve of firing rate versus distance from the preferred boundary showed a clear gradually decreasing pattern (Fig 3C, blue curve), whereas no clear tuning pattern was visible for distance from a boundary in the orthogonal direction ($\Theta_{max} + 90^o$, yellow curve). Similar results are presented in panels D-F, G-I, and J-L and correspond to the cells presented in Fig 2E–2H, 2I–2L and 2M–2P, respectively.

Furthermore, we tested whether the activity of the boundary vector cells is modulated by the allocentric swimming direction of the fish. Fig 4 presents the result of this analysis for the cells in Fig 2, respectively. We show each cell's heatmap of firing rate to position, bisected into the periods during which the fish swam towards its preferred boundary direction ($\Theta_{max} \pm 90^o$; Fig 4A, 4D, 4G and 4J, left panels, see Materials and methods) or away from its preferred boundary direction ($\Theta_{max} \pm 180^o \pm 90^o$; Fig 4A, 4D, 4G and 4J, right panels). In addition, we show the tuning curves of firing rate to the distance from the preferred boundary of each cell while swimming towards it (Fig 4B, 4E, 4H and 4K, blue curve; see Materials and methods) and away from it (yellow curve). To assess whether this analysis of allocentric swimming direction is biased by the fish's swimming speed, we also present the speed distribution along distance from the preferred boundary in each of the allocentric swimming directions (Fig 4C, 4F, 4I and 4L).

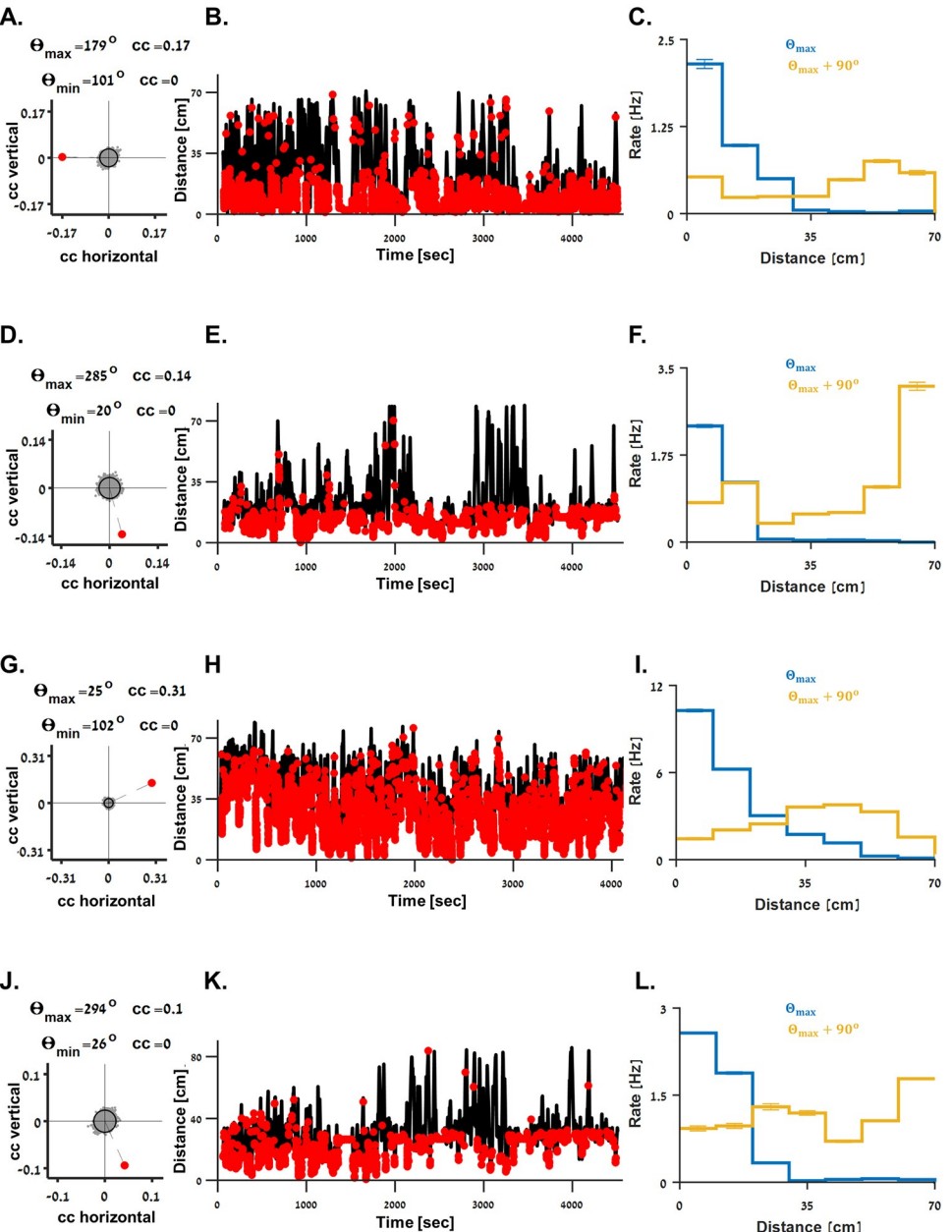

**Fig 3. Spatial characteristics of boundary vector cells. (A-C)** Spatial tuning properties of the cell presented in Fig 2A-2D. **(A)** 2D correlation coefficient of firing rate and the fish's position of the cell (red dot) and 5,000 shuffled spike trains (gray dots, the 97.5 percentile is depicted), suggesting this cell had a left boundary tuning. Preferred and null tuning directions ($\Theta_{max}$ and $\Theta_{min}$, respectively; see Materials and methods) and correlation coefficients (*cc*) are indicated. **(B)** Spiking activity (red dots) superimposed on the distance of the fish (black curve) from the preferred boundary. **(C)** Tuning curves (mean ± standard deviation) of the cell's firing rate to the distance of the fish from the boundary in the preferred direction ($\Theta_{max}$, blue curve) and its orthogonal direction ($\Theta_{max}+90^o$, orange curve). A gradually decreasing firing pattern is shown for the $\Theta_{max}$ direction. **(D-L)** The spatial tuning properties of the boundary vector cells presented in Fig 2E–2H, 2I–2L and 2M–2P, respectively. The underlying data supporting all panels in this figure can be found in a file named **Fig 3_data.mat** (see Data Availability).

For three of the examples (Fig 4D–4L), the bisected heatmaps were similar (Fig 4D, 4G, and 4J), but the firing rate was attenuated while swimming away from the preferred boundary. This is shown in the bisected tuning curves (Fig 4E, 4H and 4K, modulation ratio indices are

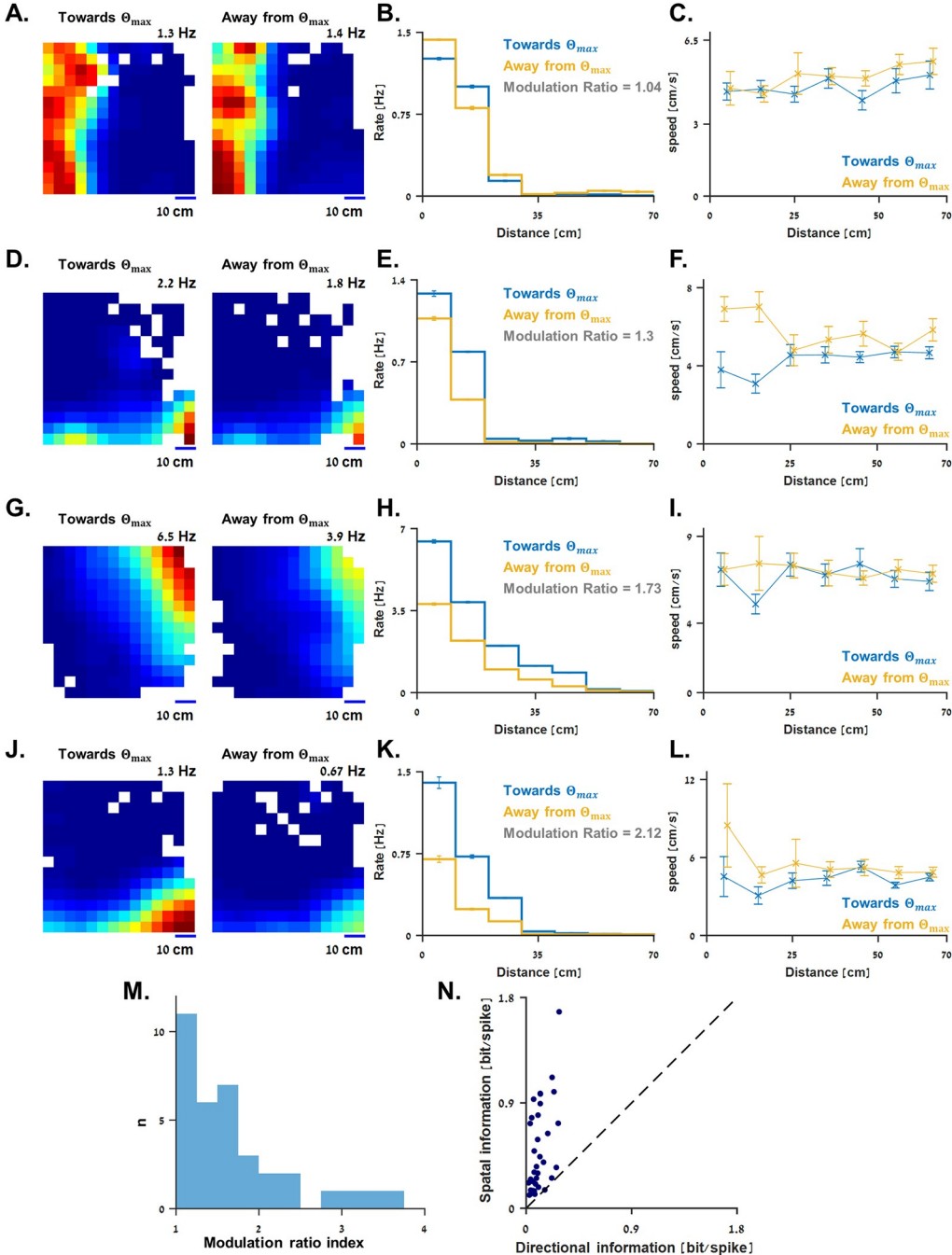

**Fig 4. Boundary vector cells tuning to allocentric swimming direction. (A)** Rate maps of the cell in Fig 2A–2D calculated solely with the periods during which the fish swam towards the preferred boundary direction (left panel) and away from it (right panel; see Materials and methods). The same color code was used for both maps. **(B)** Corresponding tuning curve of firing rate to distance from the preferred boundary calculated solely with the periods during which the fish swam towards the preferred boundary direction (blue curve) and away from it (yellow curve). Modulation ratio index (see Materials and methods) is indicated. **(C)** Distribution of swimming speed (mean ± standard deviation) along distance from the preferred boundary for the cell in A, bisected into periods during which the fish swam towards the preferred boundary direction (blue curve) and away from it (yellow curve). **(D-L)** Allocentric swimming direction tuning properties of the boundary vector cells presented in Fig 2E–2H, 2I–2L and 2M–2P, respectively. **(M)** Modulation ratio index (see Materials and methods). These values describe the strength of modulation caused by the fish's allocentric swimming direction on boundary vector cells. **(N)** Spatial information is greater than the corresponding directional information (see Materials and methods) for all boundary vector cells. The underlying data supporting all panels in this figure can be found in a file named **Fig 4_data.mat** (see Data Availability).

indicated; see Materials and methods). The additional cell shows no tuning to allocentric swimming direction (Fig 4A–4C).

To estimate the strength of the rate modulation, we calculated a modulation ratio index for all 35 boundary vector cells (see Materials and methods). This value indicates the unsigned ratio (i.e., stronger divided by weaker) between the tuning curves of firing rate to distance from a boundary in the preferred direction while swimming towards it versus away from it (e.g., the blue and yellow tuning curves in Fig 4B, respectively). Thus, a modulation ratio index around 1 indicates no clear difference between the tuning curves. The modulation ratio indices of all boundary vector cells are shown in Fig 4M, distributed in the range of [1, 3.6] with a mean ± standard deviation of 1.67 ± 0.67.

To address the possibility that our results might be mediated by a difference in swimming speed towards the boundary versus away from it, we used a $2 \times 2$ ANOVA (see Materials and methods). Under this approach, in only 2 of the 35 cells, we found a clear effect of direction on speed (ANOVA F-value in the range 11.8 to 20.89). For 23 of the cells, we found similar swimming speeds in the two directions (ANOVA F-value in the range 0 to 0.91), and the remaining 10 cells had a speed–direction relationship that was neither clearly different nor very similar (ANOVA F-value in the range 1.04 to 11.6).

Although this analysis supports the suggestion that there is a directionality tuning in the activity of the boundary vector cells in fish, it is still hard to confidently differentiate the effect of allocentric swimming direction from other behavioral aspects in space. Therefore, further investigations are needed to establish this point.

Last, in both fish and rodents, position was more important than direction: As was shown in rats [12], we calculated the spatial information carried by the goldfish boundary vector cells to be 5 times greater on average than the corresponding directional information (0.5 ± 0.37 bits/spike versus 0.1 ± 0.07 bits/spike, respectively; mean ± standard deviation; see Materials and methods).

## Boundary tuning while changing environmental geometry

To test the hypothesis that the recorded cells were tuned to a preferred boundary rather than to a specific place field in the environment, we conducted an additional experiment. In this control experiment, we measured the neural activity of the same cells before and after we changed the tank geometry. This was done by adding an additional horizontal half-wall (control in 8 of the 35 boundary vector cells). Examples of these cells are presented in Fig 5. Each cell was recorded for a full recording session (60 to 75 min; Fig 5A), after which the fish was blindfolded, and a Perspex shelf was inserted into the water tank (Fig 5B). Another full recording session was then initiated.

In one example, in the first recording session, the cell's neural activity gradually decreased with distance from the bottom of the water tank (Fig 5C). After adding the shelf (gray mark in Fig 5D), the cell's rate map (Fig 5D) shows it responded to both the bottom of the water tank and the Perspex shelf, as expected from a boundary vector cell. Additional examples showing firing patterns of boundary vector cells before and after the geometric change in the environment are presented in Figs 5E–5H and S3A–S3G.

## Population analysis of boundary vector cells

The firing patterns observed in the recorded population of boundary vector cells mostly exhibited a gradual monotonous decreasing pattern with distance from a boundary found at the preferred direction of each cell ($\Theta_{max}$; see Materials and methods), that is regardless of the distance of the fish from a boundary found in the orthogonal direction ($\Theta_{max} \pm 90^{o}$). Therefore,

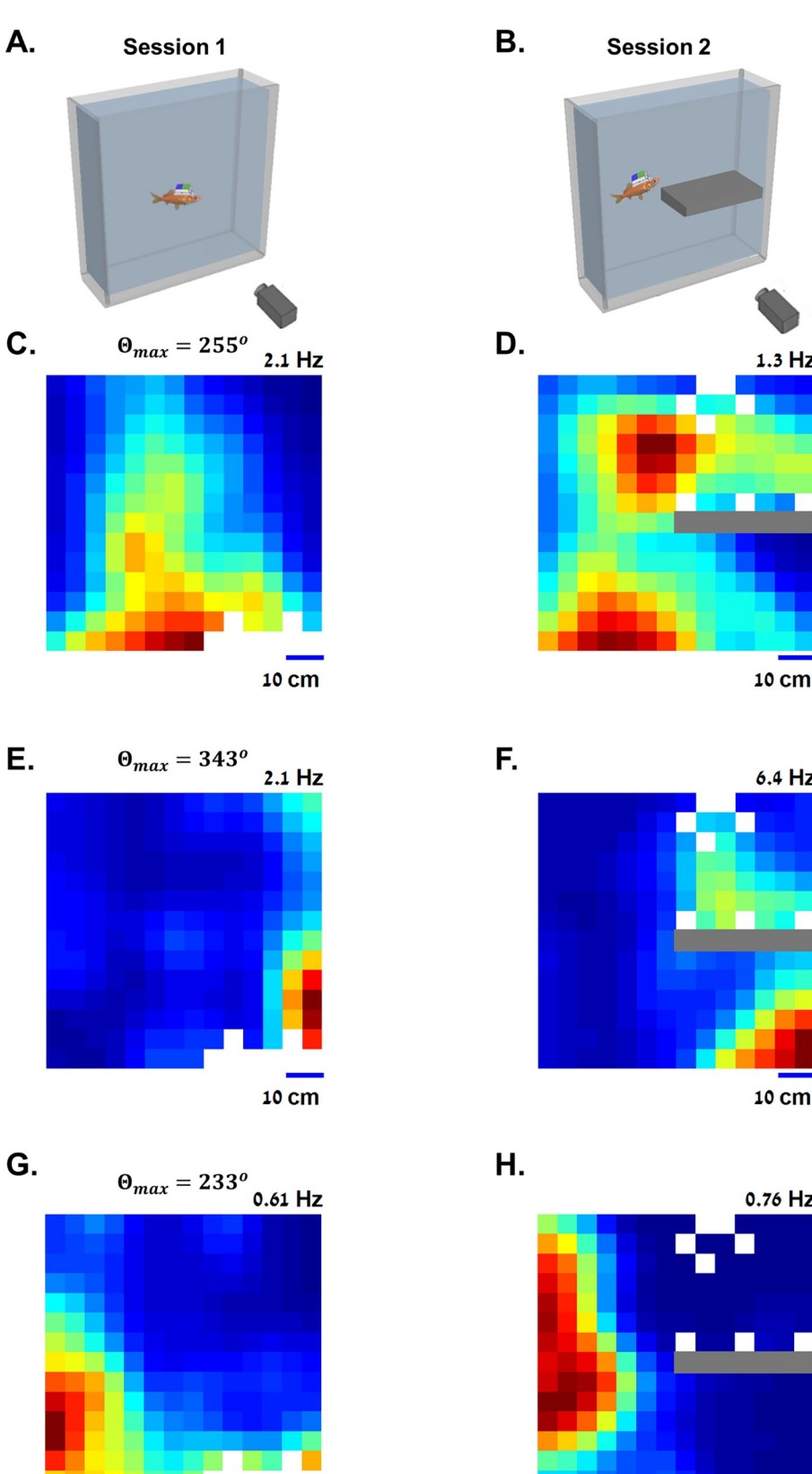

**Fig 5. Changing environmental geometry.** (**A**) The first session was recorded in the main experimental water tank. (**B**) Before the second recording session, a shelf was inserted into the water tank to modulate the geometry of the environment. (**C**) Example rate map of a boundary vector cell tuned to distance from the bottom of the water tank, color coded from dark blue (zero firing rate) to dark red (maximal firing rate, indicated on the top right side of the panel). (**D**) Rate map over the water tank of the same cell after adding the shelf. Firing was modulated by both the bottom of the water tank and above the shelf. (**E-H**) Additional examples of boundary vector cells before and after the geometric change in the environment. The underlying data supporting all panels in this figure can be found in a file named **Fig 5_data.mat** (see Data Availability).

we calculated the receptive field size (Fig 6A) of these cells as the width of the sigmoid curve, which best fit the data (see Materials and methods and S4 Fig). The results showed that these widths distributed around 67% ± 53% (mean ± standard deviation) of the width of the water tank (Fig 6A, Units: 1 [Aq] = 70 [cm]). This fit differs from a simple border cell, which is expected to have sharp sigmoidal tuning near a specific boundary in the environment [25]. It also differs from the edge cells found previously in goldfish [14], which fired in short distance from all four walls of a shallow environment. Specifically, none of cells presented here could passed these criteria for edge encoding as in [14]. Altogether, this suggests that the cells presented here can be best described as boundary vector cells, each of which is tuned to a specific boundary direction and gradually encodes the distance from it. The $\Theta_{max}$ directions across the recorded population were quasi-uniformly distributed in space (Fig 6B, color-coded by the cells' max-p values; see Materials and methods).

The characteristics of the entire recorded population are presented in Fig 6C–6G using the same color-code as in panel B. Y-axis in panels C-F is the rank-order max-p of each cell (see Materials and methods), so that the cells with the strongest spatial tuning are at the top of each panel. Rank-order correlation coefficient (r) and $p$-values are indicated to the bottom right of each one of these panels. As expected from space encoding cells, stronger spatial tuning (i.e., smaller max-p value) was characterized by a stronger in-session spatial-stability correlation-coefficient (Fig 6C, r = 0.67, $p < 1 \times 10^{-3}$), stronger spatial coherence (Fig 6D, r = 0.82, $p < 1 \times 10^{-3}$), and stronger spatial information (Fig 6E, r = 0.37, $p < 1 \times 10^{-3}$). In addition, the strength of the spatial modulation was also correlated with the mean firing rate during the entire recording session (Fig 6F, r = 0.45, $p < 1 \times 10^{-3}$).

Furthermore, the mean action potential properties of each cell (amplitude versus width at half maximum for the electrode that recorded the strongest spikes) are presented in Fig 6G. We used a two-sample $t$ test and found that the ratio between the peak spike amplitude and its width can be distinguished between the 35 boundary vector cells and the rest of the recorded population ($p < 1 \times 10^{-3}$, t = 3.57 and degrees of freedom = 117).

For the 35 boundary vector cells, we also show the mean firing rate in the 30% strongest bins of each rate map (first half versus second half of each experiment; Fig 6H). A paired-sample $t$ test ($p = 0.011$, t = 2.67 and degrees of freedom = 32) suggested the firing rate in the receptive field of the boundary vector cells might decrease over time. This might be related to adaptation of the fish to the environment, but further study is needed to establish this point. Last, we also show a lack of relationship between the preferred boundary direction of these cells and their peak firing rates (Fig 6I; circular-linear correlation-coefficient, r = 0.25 and $p = 0.14$).

The boundary vector cells' receptive fields covered the entire water tank. A rate map overlap of the top 50 percentile bins of all boundary vector cells is presented in S3H–S3J Fig (divided into 3 different panels by the preferred direction solely for ease of visualization). The full coverage suggests that boundary vector cells might be sufficient for the fish to encode position in small environments.

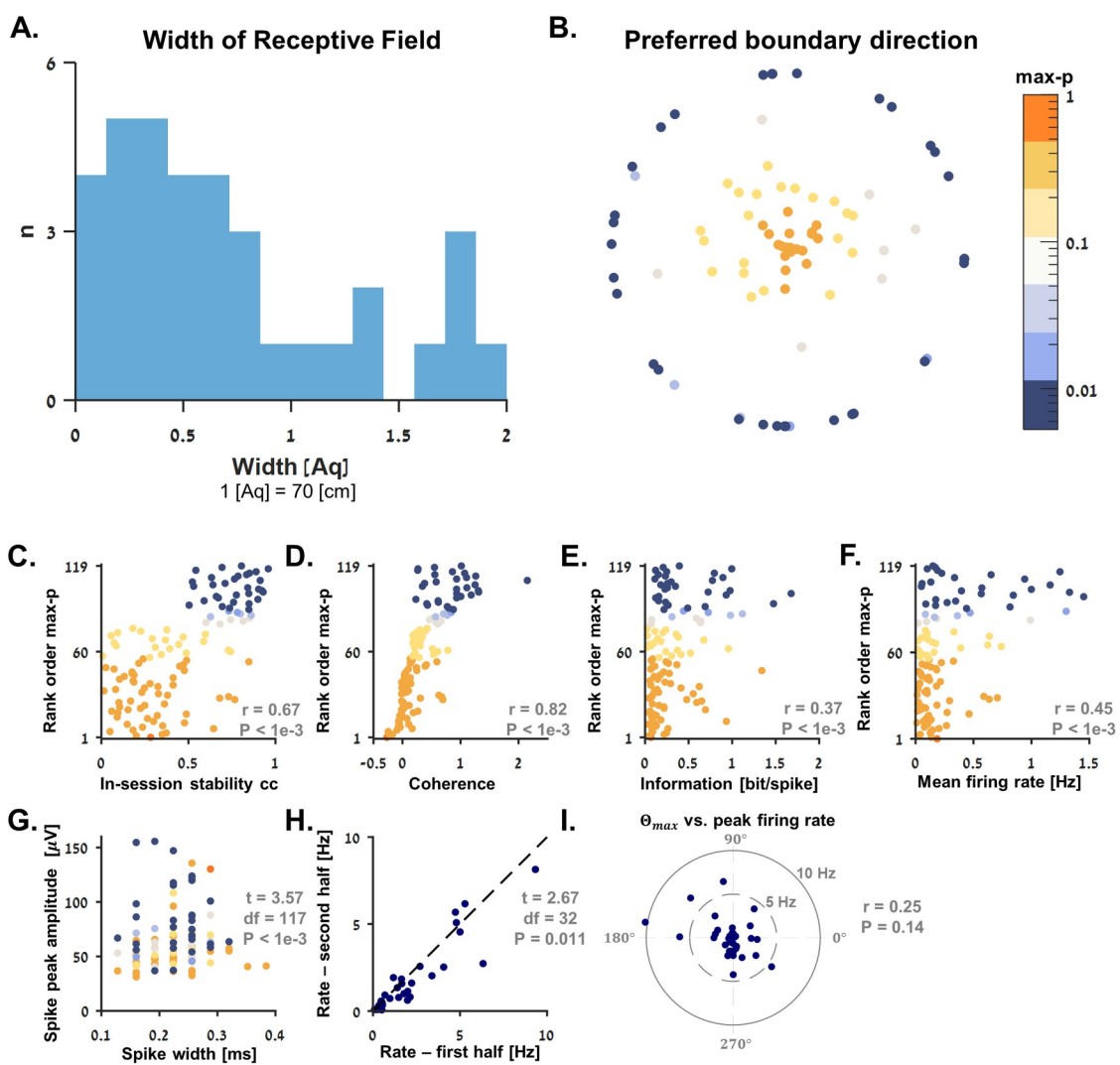

**Fig 6. Population analysis of boundary vector cells. (A)** Distribution of receptive field widths (i.e., the width of the sigmoid curves fitted to the boundary vector cells; see Materials and methods) relative to the width of the water tank. Units: 1 [Aq] = 70 [cm]. **(B)** The population max-p values and preferred boundary direction $\Theta_{max}$ (see Materials and methods) of all 119 single units recorded from the brains of 15 fish. The dots are distributed on a radial plot where $Radius = log\left(\frac{1 - max\_p}{max\_p}\right)$ folded to the range [0, 5] and $Angle = \Theta_{max}$. The color bar on the right-hand side of panel B spans the population max-p values on a logarithmic scale and shows the distribution of cells in panels B-G. **(C-G)** Population statistics. Y-axis in panels C-F is the rank-order max-p value of each cell (see Materials and methods), so that the cells with the strongest spatial modulation are in the top of each panel. Statistics summary of these panels is indicated in the bottom right side of each panel. For each cell, we show the following properties: **(C)** in-session stability correlation coefficient, **(D)** spatial coherence, **(E)** spatial information, **(F)** mean firing rate, and **(G)** mean action potential amplitude and full width at half maximum for the electrode recorded the strongest spikes. Two-sample $t$ test results are indicated. **(H)** Mean firing rate in the 30% strongest bins in the rate maps of the 35 boundary vector cells in the first vs. second half of the experiments. Paired $t$ test results are indicated. **(I)** Preferred boundary direction ($\Theta_{max}$) of the 35 boundary vector cells vs. peak firing rate. The angle in the polar plot is $\Theta_{max}$, and the radius is the peak firing rate. Circular-linear correlation coefficient (r) and $p$-value are indicated. The underlying data supporting all panels in this figure can be found in a file named **Fig 6_data.mat** (see Data Availability).

We wanted to verify that the spatial firing pattern of these cells is, indeed, due to position encoding rather than other aspects of locomotion, such as swimming direction or swimming speed. For example, a gradual distribution in space of swimming speed could alone explain the observed phenomenon. Therefore, we controlled for these aspects of locomotion across the water tank to make sure no clear gradual pattern emerged (S5 Fig).

## Beta oscillations in boundary vector cells

About half of the boundary vector cells exhibited rhythmic neural activity. An example of one such cell is presented in Fig 7A–7C. This cell is a boundary vector cell with a gradually decreasing firing pattern with distance from the bottom of the water tank (Fig 7A). Examining the histogram of interspike interval of this cell (Fig 7B) revealed a periodic spacing pattern between the peaks of the histogram. This was further manifested in the frequency domain: After the histogram was normalized by the total number of spikes, we calculated the power spectral density function of the normalized histogram (Fig 7C), where a local maximum appeared at around 16Hz; in other words, this neuron oscillated rhythmically in the low-beta frequency range. A counter-example of a cell that was not tuned to space is presented in Fig 7D–7F.

Examining the position and magnitude of the peak power spectral density in the beta range of the entire population (Fig 7G, color-coded by the max-p values; see Materials and methods) revealed a clear connection between the beta oscillations and the boundary vector cells in the central telencephalon of the goldfish. To estimate the properties and prevalence of the beta oscillations in the population, we used a threshold of 0.0015 dB/Hz for the peak power of the spectral density. The findings showed that 16 of the 35 boundary vector cells (approximately 46%) and 1 out of the other 84 cells crossed this threshold with a peak spectral density in $15.25 \pm 1.62$ Hz (mean ± standard deviation). Different thresholds were tested to verify that these results were independent of the chosen threshold (Fig 7H; see Materials and methods).

## Sigmoidal firing patterns in space

To further characterize the spatial firing patterns observed in the boundary vector cells, we tested two tuning models: (1) Sigmoid tuning along the cell's preferred direction, resembling distance encoding; and (2) Gaussian tuning in the form of a 2D Gaussian centered in proximity to the environmental boundaries, resembling local encoding. We fitted the two models for the firing rate map of each cell and tested the goodness of fit for the two models.

This process was first validated using a simulated dataset (examples are presented in S4 Fig; see Materials and methods). Using the simulated data, we set the thresholds (S4J and S4K Fig; see Materials and methods) for classifying a cell as distance encoding or Gaussian encoding (as in a place cell). As expected, the results showed that most boundary vector cells fit better into the sigmoid model than the Gaussian model (see S1 Table). This emerged in the correlations between the data and the models (S4K Fig).

Since a 2D Gaussian with a very broad distribution on only one axis is similar to a sigmoid distribution, these two models are not mutually exclusive. Nevertheless, this analysis helped us to rule out the hypothesis that the cells we recorded were a subset of other typical locally activated space-encoding cells found in vertebrates, characterized by a narrow Gaussian distribution that happen to be centered near boundaries.

## Stability of the analysis

To ensure the validity of our analysis over an inhomogeneous swimming trajectory across the water tank (S6 Fig), we ran our analysis pipeline on a set of simulated data. The simulated cells' activity was drawn from three different spatial tuning curve types (500 shuffled spike trains for each tuning curve). We used a fish's recorded trajectory, which did not cover the water tank homogenously (corresponding to the trajectory of the cell presented in S2D Fig). Tuning curves of firing rate to position were chosen from three different fish. They were gradually decreasing (S6A Fig, right panel), quasi-uniformly distributed (S6B Fig, right panel), or gradually increasing (S6C Fig, right panel) with the position of the fish along the vertical axis of the water tank (see Materials and methods).

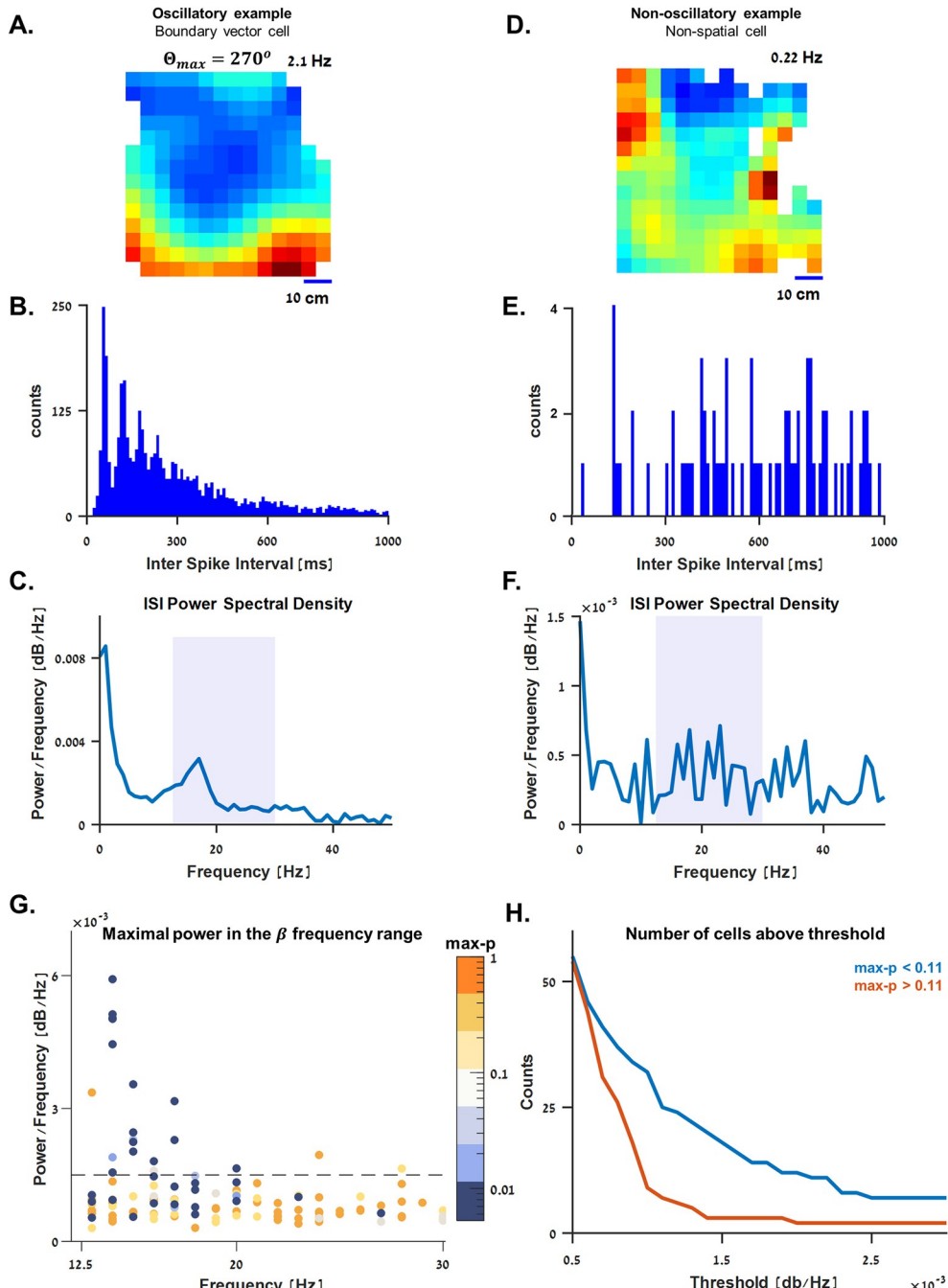

**Fig 7. Beta oscillations in boundary vector cells. (A-C)** Example of a boundary vector cell with a periodic interspike interval (ISI) pattern. **(A)** Firing rate heatmap of a boundary vector cell tuned to the bottom of the water tank, color coded from dark blue (zero firing rate) to dark red (maximal firing rate, indicated on the upper right side of the panel). **(B)** ISI histogram of the cell in A. Even spacing between the peaks suggests periodic oscillations of the neural activity. **(C)** Power spectral density of the histogram in B (normalized). Colored background marks the typical frequency range of beta waves (12.5–30 Hz). A local peak is shown at approximately 16 Hz, suggesting that this cell exhibited beta-rhythm oscillations. **(D-F)** A counter-example of a cell with no clear spatial tuning and no specific pattern in the ISI histogram. **(G)** Maximal power spectral density in the beta waves range for the entire population. The color bar spans the population's max-p values (see Materials and methods) on a logarithmic scale. As shown, roughly half of the boundary vector cells (blue dots) exhibited beta rhythm oscillations at 15.25 ± 1.62 Hz (mean ± standard deviation). **(H)** Different thresholds were tested to estimate the prevalence of beta oscillations in the recorded population (see Materials and methods). The median value of the population's max-p was used to divide the data into groups similar in size. Regardless of the threshold tested, the cells that were more spatially tuned (i.e., below median-max-*p* = 0.11, blue

curve) were more abundant above the threshold than others (i.e., above median-max-p value, orange curve). The underlying data supporting all panels in this figure can be found in a file named **Fig 7_data.mat** (see Data Availability).

For the decreasing firing pattern, 499 of the 500 simulated cells were indeed classified as spatially modulated. This was also true for 498 out of the 500 cells obtained for the increasing firing pattern (false positive rate <0.004; S6D Fig). None of the cells obtained from the quasi-uniform firing rate (S6D Fig) were classified as spatially modulated (false negative rate <0.002). This confirmed that the analysis for spatial tuning was not sensitive to occupancy correction artifacts in the firing rate map due to partially visited parts of the water tank.

## Discussion

We found evidence for spatially modulated cells in the central area of the goldfish telencephalon. For most cells, the firing pattern gradually decreased with distance from a boundary in the cell's preferred direction. For others, peak firing rate was at a short distance (up to 20 cm) away from the boundary. These firing patterns could be tuned to boundaries in all directions of space. Specifically, these cells could be tuned to the bottom of the water tank, the water's surface, the water tank's walls, or salient features of the environment such as corners or an additional boundary. Furthermore, the gradual firing pattern of each cell usually spanned a large portion of the experimental environment.

The relatively large receptive fields of the cells reported here differentiate them from the typical mammalian border cells in which the firing rate decreases sharply near one or more environmental boundaries [25]. The large receptive fields also differ from the previously reported edge cells observed in the goldfish Dlv region in a shallow water tank [14]. The previously published edge cells were also activated near all the environmental boundaries, unlike the specific tuning to a preferred boundary or even part of a boundary presented in this study.

Altogether, the cells presented here mostly resemble the mammalian boundary vector cells found in the rat subiculum [12]. These mammalian pyramidal cells were shown to exhibit gradually decreasing activity with distance from an allocentric boundary and were argued to be an integral part of place encoding. It has been argued that boundary vector cells may have a larger receptive field than a typical border cell [12], which is in line with the results presented here.

However, some differences can be observed when comparing the population shown here to the mammalian boundary vector cell population. Unlike the clear omnidirectional tuning shown in mammals [12], it is suggested here that part of the boundary vector cells in the goldfish may be also sensitive to the allocentric swimming direction of the fish. This might imply the difference between the encoding model for boundary distance used in the different neural systems. Nevertheless, in both cases, the spatial information was about 5 times greater than the directional information carried by the boundary vector cells (Fig 4N; see also [12]). Additional studies in different environment geometries, including in a cylindric environment (as was used in [12]), might help decipher whether these differences do exist. Furthermore, they might help to establish the dependency between allocentric swimming direction and firing rate of boundary vector cells in fish.

Another difference observed in fish was neural oscillations in the low-beta range (13–20 Hz). We found these oscillations in the activity of many boundary-vector cells, and we found almost no oscillations in cells, which were not modulated by position. To the best of our knowledge, there are no previous reports of correlations between spatial localization and beta oscillations. Neural oscillations associated with spatial memory are often reported to be in the

theta frequency range [19]. Thus, these findings may form an essential aspect of the comparative approach of studying spatial encoding across species and support the novelty of spatial encoding in fish and the importance of rhythmic oscillations for spatial encoding in general.

The differences between these neural systems in fish and rodents may be related to the natural characteristics of these animals. While rodents are terrestrial animals navigating in quasi-2D environments, fish constantly navigate in a 3D aquatic world, making the sensory cues they use for navigation very different. Therefore, we believe that additional experiments, including sensory cues variations, are needed to fully understand the neural basis of boundary vector cells in fish.

Specifically, one interesting variation has to do with hydrostatic pressure. The steep pressure gradient fish experience in the water's vertical dimension may give them direct information about their position, and its derivative sign gives immediate information about the allocentric swimming direction in that dimension. Therefore, a subset of the cells presented here may reflect depth encoding. Several studies have suggested that fish can sense hydrostatic pressure in the telencephalon and use it for navigation [26–36]. However, additional experiments are needed to show depth encoding per se. For example, neural activity could be recorded in an environment that changes hydrostatic pressure in an invariant visual scene.

The findings presented here contribute to defining the basic inventory of spatial and kinematical cells in the goldfish telencephalon. The abundance of spatially modulated cells in the central part of the telencephalon underscores the importance of this brain region for navigation. For these reasons, the data reported here constitute an essential step towards a better understanding of the representation of space in the brains of fish, the largest group of vertebrates. Using a comparative approach to the well-studied mammalian navigation system can thus help decipher which parts of this system have evolved in different taxa and how.

## Materials and methods

### Experimental model

A total of 15 goldfish (*Carassius auratus*) 13 to 18 cm in length and weighing 100 to 250 g were used in this study. Each fish was housed in a home water tank at room temperature and brought to the experimental water tank for the recording sessions. The room was set to a 12/12-h day–night cycle under artificial light. All the experiments were approved by the Ben-Gurion University of the Negev Institutional Animal Care and Use Committee and were in accordance with government regulations of the State of Israel.

### Wireless electrophysiology

The implant preparation and transplanting process, as well as the behavioral electrophysiology details, are fully described in our previous publications [37,38]. Briefly, extracellular recordings were obtained from the goldfish's central telencephalon using 3 or 4 tetrodes. The data were logged on a small recording device (Mouselog-16, Deuteron Technologies, Jerusalem, Israel) placed in a waterproof case mounted on the fish's skull. In addition to the tetrodes, a reference electrode was placed near the brain to detect possible motion artifacts rather than neural activity. The tetrodes were moved in the brain between recording sessions using the built-in microdrive of the implant. Wireless communication via a PC and a transceiver (Deuteron Technologies, Israel) was used to control and synchronize the data logger. The neural signals were recorded at 31,250 Hz and band-pass filtered at 300 to 7,000 Hz. A colored styrofoam marker attached to the waterproof case was used to neutralize the buoyancy of the implant (i.e., total average density of 1 g/cm$^3$) and to detect the fish's position and swimming direction on the video recordings.

## Surgery and stereotaxic procedure

During surgery, the anesthetized fish was placed in a holder on the operating table and perfused through its mouth with water and anesthetic (MS-222 200 mg/l, NaHCO3 400 mg/l 1, Cat A-5040, and Cat S-5761, Sigma-Aldrich, USA). All surgery specifics are described in detail in [38]. We targeted the central area of the telencephalon by taking the anterior mid margin of the posterior commissure as the zero point for the stereotaxic procedure [39]. Then, using a mechanical manipulator, we moved the tetrodes 1 mm posteriorly and 1.5 mm ventrally. We used the microdrive attached to the tetrodes to further adjust them in the dorsal/ventral axis after surgery.

## Experimental water tank

The experimental water tank was 0.7 m in the horizontal axis and 0.7 m in the vertical axis. The third axis was foreshortened (0.2 m), creating a quasi-2D environment (Fig 1A). The water level during the experiments was constant—0.7 m.

The walls on 3 sides of the tank (one horizontal side and the two short, degenerated sides) were covered and blocked the visual cues from the room. The water tank faced a corner of a room, and the only visual cues from the outside came from the uncovered horizontal side, including a Raspberry Pi and its small camera facing the center of the tank, and a PC (not in front of the tank) communicating with the neural logger (PC screen was covered during the experiments). The top of the water tank was not covered thus ceiling was also visible for the fish.

## Video recording

A Raspberry Pi camera (Raspberry Pi 3B microprocessor) oriented towards the center of the water tank was used to localize the fish in the vertical and horizontal dimensions. To synchronize the neural activity and the video recordings, we used Arduino Uno, which sends simultaneous TTL pulses to the data logger's transceiver and the Raspberry Pi unit. Each recording session lasted 60 to 75 min while the fish navigated freely in the water tank.

## Histology

The brains of 11 out of 15 fish were fixed in 4% paraformaldehyde overnight (Electron Microscopy Sciences, CAS #30525-89-4) and then were immersed in a 40% glucose solution for cryoprotection. After freezing, the brains were cryo-sliced (40 μm slices) and Nissl stained to reveal the position of the electrodes in the brain (i.e., the central area of the telencephalon). All slices were then scanned by automated microscope (Panoramic scanner, 3DHISTECH, Hungary) with a 20× (NA 0.8) and a 40× (NA 0.95) objective. The slices were compared to the neuroanatomical landmarks of [40] to confirm the recording region.

## Spike sorting

The raw neural data were band-pass filtered at 300 to 7,000 Hz. Then, a threshold was set to detect the action potential timings. Manual single cell clustering was done for each tetrode separately by standard spike sorting methods [41,42], including PCA analysis of the spike amplitudes, widths, and waveforms across all four electrodes. Units not clearly separated in the PCA space and units unstable over time were eliminated from further analysis. In addition, spikes that appeared simultaneously in more than one tetrode or reference electrode were considered motion artifacts or electronic noises and were removed from the analysis. Examples of clean spikes are presented in Fig 1C–1E. More details can be found in [37,38].

### Fish trajectory and firing rate-map analysis

The fish's location and orientation in each video frame were detected using the open-source pose-estimation algorithm by DeepLabCut [43]. This deep-learning algorithm detected several coordinates of the implant on the fish's head in each video frame. Next, the coordinates were translated to the vertical-horizontal position in cm units. Then, the fish's trajectory and spike positions were binned using a 5 cm × 5 cm grid and smoothed using a 2D Gaussian filter (σ = 20 cm) to obtain auxiliary maps of occupancy per bin and spike count per bin, respectively. A bin-by-bin division of spike counts by occupancy yielded the occupancy-corrected firing rate map for each cell. Examples of the steps in this correction process are presented in S7 Fig. The corrected rates in each map were color-coded from zero (dark blue) to the maximal firing rate of each cell (dark red). Bins in which the fish spent fewer than 2 seconds were discarded from the analysis and colored white in the figures.

### Spatial tuning statistics

To evaluate spatial modulation, we applied the standard statistical tests for spatial modulation in the literature: in-session stability, spatial coherence, and spatial information [44]. For the spatial information, we also used a threshold of 0.1 bits/spike.

To test for in-session stability, we calculated the firing rate maps for the first and second halves of each experiment and calculated the correlation coefficient between them. Then, 5,000 shuffled spike trains were obtained by calculating the interspike intervals (ISIs), shuffling the ISIs by random permutation, and using the cumulative sum to obtain a shuffled spike train. For each shuffled spike train, we created a heatmap using the trajectory of the second half of the experiment and calculated the correlation coefficients between the shuffled heatmap and the heatmap of the first half of the experiment. Comparing the shuffled results with the data yielded a $p$-value for stability representing the probability of obtaining greater in-session stability than that of the cell by chance.

For the shuffled spike trains, we also calculated the spatial coherence and spatial information values and compared them to the nonshuffled cellular result to obtain two additional $p$-values. We assigned each cell a max-$p$ value (as used in Figs 6B–6G, 7G and S4K), which was the largest $p$-value of those calculated in the three tests. Cells with a max-$p < 0.025$ and for which the spatial information was at least 0.1 bit/spike were counted as spatially modulated cells. This category was used for the purposes of determining the tuning width, rate stability, and directional preferences in spatially modulated neurons and to help estimate the prevalence of beta rhythm oscillations and of boundary vector cells in the population. In the latter case, we also tested different max-p and spatial information thresholds and observed that the prevalence of boundary vector cells was not strongly affected (see S2 Table).

### Directional preferences

As one measure of neuronal tuning, we calculated the correlation coefficient between the cellular firing rate and the position in both the vertical and horizontal axes of the water tank. We used a shuffle analysis to get a perspective on the magnitude of these correlations. Specifically, 5,000 shuffled spike trains were obtained by ISI shuffling as described above. For each shuffled spike train, the correlation coefficients were calculated as with the original data. The results of this analysis were used solely for visual assessment (Fig 3A, 3D, 3G and 3J), whereas the actual analysis of the strength of the correlation was taken from the correlation in the preferred and orthogonal directions as defined below.

To determine the preferred boundary direction of the boundary vector cells, we calculated the correlation coefficients between the firing rate and projection of position along 180 axes evenly spaced on the half circle and determined the one for which the value of the correlation coefficient was maximal ($\Theta_{max}$, the preferred boundary direction) and the one for which the

absolute value of the correlation coefficient was minimal ($\Theta_{min}$). As expected, $\Theta_{min}$ was tightly distributed around the orthogonal direction of $\Theta_{max}$. To simplify the analysis, we used $\Theta_{max} \pm 90^o$ as the null direction for all neurons.

Then, we calculated the tuning curve of firing rate versus distance from the boundaries found at the $\Theta_{max}$ and $\Theta_{max} \pm 90^o$ directions (as is Fig 3C, 3F, 3I and 3L). This was done by dividing the projection of spike locations on the specified axis by the occupancy using bins of 10 cm. Error bars were calculated by repeating this process 1,000 times with only 50% of the data each time and then calculating the standard deviation in each bin.

## Allocentric swimming direction tuning analysis

To test how allocentric swimming direction affected the activity of boundary vector cells, we bisected the data using the displacement direction of the fish. Thus, displacement direction in the range $\Theta_{max} \pm 90^o$ was considered swimming towards $\Theta_{max}$, and else was considered swimming away it. We used this bisection of the data to plot the heatmaps, tuning curves, and speed distributions shown in Fig 4A–4L.

To quantify the effect of the allocentric swimming direction, we calculated a modulation ratio index (Fig 4M); namely, the ratio that minimizes the least square error between each cell's two tuning curves—firing rate versus distance from the preferred boundary while swimming towards it and while swimming away from it (e.g., blue and yellow curves in Fig 4B, respectively). For simplicity, this ratio was unsigned, i.e., the stronger tuning curve was divided by the weaker for all cells.

Since behavior might differ while swimming either towards a boundary or away from it, and as an additional measure of stability of this analysis, we calculated the swimming speed distributions of all boundary vector cells along the distance from the preferred boundary for the two bisected data sets (examples are presented in Fig 4C, 4F, 4I and 4L).

Further, we wanted to assess statistically whether the modulation ratio effect might be partially mediated by different swimming speeds towards the boundary versus away from it. Therefore, we used $2 \times 2$ ANOVAs for each fish with allocentric swimming direction (towards versus away) and distance from the wall (10 cm or 20 cm, i.e., in each cell's receptive field) as independent variables and speed as the dependent variable. To avoid time correlations in the dependent variable, we down sampled the data to at least 3 seconds between samples: the auto-correlation of the speed goes to 0 at this time difference. This gave us on average 94 independent data points (range: 22 to 322) for the ANOVA for each fish. We determined that a fish swam faster in one direction compared to the other it in case the Bonferroni-corrected $p$-value of the main effect of direction on speed was at $p < 0.05$. We determined that the fish swam nearly the same speed towards and away from the boundary if the main effect of direction had a $p > 0.15$. For other $p$-values between these two ranges, we classified the effect of direction on speed as indeterminate.

Last, to estimate the relationship between the directional and spatial information in boundary vector cells (Fig 4N), we repeated the process described in [12] to calculate the mutual information between firing rate and location and compared it to the mutual information between firing rate and the allocentric swimming direction of the fish. To do a fair comparison, we used exactly 48 bins (7.5 degrees each) to calculate the directional information and 48 bins on average for the spatial information.

## Varying geometry experiment

To test the hypothesis that cells that are tuned to boundaries are also tuned to an additional unfamiliar boundary, in some of the experiments, we altered the geometry of the experimental

environment. This was done by placing an additional boundary—a nontransparent horizontal shelf (0.35 m in size), about 0.35 m deep near the right wall of the experimental water tank (see Fig 5B for schematics of the altered environment). This manipulation was done between two subsequent recording sessions while the fish was blindfolded. We recorded 19 of the 119 cells reported in this study in the two different environments.

## Beta oscillations analysis

To test for rhythmic oscillations in the beta frequency range, we calculated the histogram of the ISI of each cell using bins of 10 ms. Then, we normalized the histogram and calculated the power spectral density function of the normalized histogram with bins of 1 Hz. For each cell, we calculated the frequency at which the power spectral density was maximal in the beta wave range (12.5 to 30 Hz). Cells with a maximal power spectral density greater than 0.0015 dB/Hz were labelled oscillatory to estimate the prevalence of beta oscillations in the population of boundary vector cells.

To verify that choosing 0.0015 dB/Hz as a threshold did not strongly affect the results, we tested different thresholds (Fig 7H). We used the median-max-p value of the population (0.11) to divide the data into two groups similar in size: those with a max-p value above 0.11 (i.e., with a relatively weak spatial tuning, orange curve) and those with a max-p value below 0.11 (i.e., with a relatively strong spatial tuning, blue curve). The results suggested that regardless of the threshold used, the cells tuned to space (i.e., below the median-max-p value) were more abundant above the different thresholds.

## Distance versus local encoding test

Since it remained unclear whether there are "place cells" in the fish telencephalon, and if so, what their properties are, we tested three alternative models to determine what best described the firing patterns observed for the spatially modulated cells: a sigmoid model assuming boundary-vector modulation (i.e., a gradual code along the cell's preferred direction and a uniform code along the orthogonal direction), a Gaussian model assuming local modulation (as in place cells), and a uniform model of no modulation (control). We tested our ability to differentiate these alternatives on simulated data (S4A–S4J Fig). Each model was tested on a dataset simulated with three different trajectories and three different tuning curves of firing rate to position as recorded in our data.

For the sigmoid model, we simulated 500 gradual firing patterns generated from the three tuning curves of firing rate to position. These were rotated to random preferred directions in space, while in the null (orthogonal) direction, the tuning of rate to position was bounded to a uniform distribution. An example is presented in S4A Fig. Each simulated firing pattern was then fitted with a 2D Gaussian (S4B Fig) and with a sigmoid (S4C Fig).

We used the Matlab **fit** function to minimize the least square difference from the sigmoid:

$$Sigmoid(L, U, x_{mid}, \lambda, x) = L + \frac{U - L}{1 + e^{\left(-4 \cdot log \frac{3 \cdot x - x_{mid}}{\lambda}\right)}}$$

where $L$ and $U$ are the lower and upper asymptotes of the sigmoid, respectively, $x_{mid}$ is the point where the curve goes through 50% of the transition, and $\lambda$ is the width in which 80% of the transition occurs (also used as the boundary vector cells' width of tuning shown in Fig 6A). The parameter $x$ represents the simulated distance from a boundary in the preferred direction.

The quality of the fit was assessed as the correlation coefficient between the rate map of the simulated data and the rate map generated by the fit model. Each simulated dataset was also fitted with a Gaussian using the Matlab **fitgmdist** function, and goodness of fit was assessed in the same way.

Simulated firing patterns for the Gaussian model were generated by choosing Gaussians whose centers were within 10 cm of the edges of the environment. This simulated the near border tuning present in our recordings and provided an alternative to the sigmoid tuning. The same pipeline was used for this model as for the former. An example of a simulated Gaussian encoding cell and rate maps fitted from the two models is presented in S4D–S4F Fig. We also repeated the process using 500 rate maps generated from a uniform distribution (S4G–S4I Fig).

The results of the simulation (S4J Fig) showed that the cells exhibiting a gradual firing pattern along a preferred direction could be distinguished from those exhibiting Gaussian firing patterns using the correlation coefficients between the cell's rate map and the rate maps from the fit models. For this purpose, we drew threshold lines, which created a classification criterion with very few false positives. That is, every neuron with a correlation to the sigmoid model above 0.7 and above the correlation to the Gaussian model was classified as distance encoding. Only 4 of the 500 Gaussian-encoding simulated cells and 7 of the non-encoding (uniform) simulated cells were classified in this way as distance encoding (a false positive rate of around 0.011). In addition, 36 of the 500 cells simulated with a gradual firing pattern were not classified as distance encoding (a false negative rate of around 0.072). We then applied the thresholds derived from the simulations to the real data in order to characterize the firing patterns of the actual neurons (S4K Fig).

## Supporting information

**S1 Fig. Additional histology examples.** (**A**) Fish brain structures. Red cross: Sections 1 and 2 correspond to the brain sections presented in panels B and C, respectively. (**B, C**) Examples of recording sites in the goldfish central telencephalon and the corresponding brain region (right panel, anatomical diagram based on [40]). Black x's show location where the boundary vector cells were recorded.
(TIF)

**S2 Fig. Additional examples—Boundary vector cells.** (**A**) Schematic overview of the experimental setup. Dashed circle presents the $\Theta_{max}$ axis. We used camera-east as zero with an anticlockwise progression. (**B-W**) Boundary vector cells examples. For each cell presented are the firing rate map (left panels) color-coded from dark blue (zero) to dark red (maximal firing rate, indicated) and the tuning curves (mean ± standard deviation, right panels) of firing rate to the distance of the fish from the preferred boundary. Preferred direction ($\Theta_{max}$) is indicated. (**B-I**) Vertical tuning examples ($\Theta_{max} = 90^o \pm 15^o$ or $270^o \pm 15^o$). (**J-N**) Horizontal tuning examples ($\Theta_{max} = \pm 15^o$ or $180^o \pm 15^o$). (**O-W**) Other tuning examples. The underlying data supporting panels B-W in this figure can be found in a file named **SuppFig 2_data.mat** (see Data Availability).
(TIF)

**S3 Fig. Additional examples—Changing environmental geometry.** (**A, B**) The cells presented in panels C-G were recorded first in the main experimental water tank (A). Then, before the second recording session, a shelf was inserted into the water tank to modulate the geometry of the environment while the fish was blindfolded. (**C-G**) Examples of boundary vector cell rate maps before (left panels) and after (right panels) the geometric change in the environment. Maps are color coded from zero (dark blue) to the maximal firing rate of each cell (dark red, indicated). (**H-J**) Overlap of the entire 35 boundary vector cells. Each color represents the bins in space in which one cell was firing at a rate of at least half of its maximal rate. The population covers the entire water tank. For assessment only, the population was divided into 3 different panels: (**H**) cells with a vertical tuning direction, (**I**) cells with a horizontal

tuning direction, and (**J**) cells with diagonal tuning direction. The underlying data supporting panels C-J can be found in a file named **SuppFig 3_data.mat** (see Data Availability).
(TIF)

**S4 Fig. Using simulated cells to validate the analysis and setting thresholds for recorded data.** Three groups of cells (500 cells in each group) were simulated with different tuning properties in space. For each group, two models were tested to characterize the resulting firing pattern. (**A**) Example of a simulated rate map with a distance tuning pattern. Same color bar was used for the rate maps in panels B and C. (**B**) A rate map fitted to the rate map in A using 2D Gaussian tuning. (**C**) A rate map fitted to the rate map in A using a sigmoid tuning. (**D-F**) Another simulated example with a local tuning pattern near a boundary in the environment. (**G-I**) Another simulated example with a uniformly distributed firing pattern in space. (**J**) Three groups of 500 firing patterns were simulated, and correlation coefficients were calculated for each fitting method (see Materials and methods). The three groups—distance encoding, Gaussian encoding, and non-encoding—could be classified using boundaries of 0.7 and above or below the identity line. (**K**) Correlation coefficients between recorded data rate maps and the rate maps generated by a Gaussian model and a sigmoid model (see Materials and methods). Results suggest that the boundary vector cells can be better described with a sigmoid firing pattern in space rather than a Gaussian firing pattern. Thresholds (dashed lines) were derived from the simulated dataset (panel J). The color bar on the right-hand side of panel K spans the population max-p values (see Materials and methods) on a logarithmic scale such that the blue dots represent the strongest spatially modulated cells. The underlying data supporting all panels in this figure can be found in a file named **SuppFig 4_data.mat** (see Data Availability).
(TIF)

**S5 Fig. Speed and allocentric swimming direction distribution across the water tank.** Examples of swimming speed and allocentric swimming direction distributions of the boundary vector cells presented in Fig 2. For each cell, a swimming speed map (left panels) is presented, color coded from dark blue (zero) to dark red (maximal swimming speed, indicated on the top right side of each panel). Also shown are the allocentric swimming direction map (right panels), color coded from dark blue (−pi) to dark red (+pi). No clear patterns emerged. Examples correspond to the cells presented in (**A**) Fig 2A–2D and 2M–2P, (**B**) Fig 2E–2H, and (**C**) Fig 2I–2L. The underlying data supporting all panels in this figure can be found in a file named **S5 Fig_data.mat** (see Data Availability).
(TIF)

**S6 Fig. Validating the stability of the analysis against nonuniform coverage of a fish's trajectory.** We used three tuning curves of firing rate to position in the vertical axis as recorded from three different fish to simulate firing patterns in a real trajectory (panel A, left panel, black curve), which partially covered the experimental water tank. This trajectory corresponds to the cell presented in S2D Fig. For each tuning curve, we simulated 500 spike trains. (**A-C**) Examples of spiking patterns (red dots, left panels) over the swimming trajectory (black curves) together with a color-coded occupancy-corrected heatmap (middle panels, same color bar as in panel A) and the tuning curves used to simulate them (right panels). The tuning curves either gradually decreased (**A**), were quasi-uniformly distributed in space (**B**), or gradually increased (**C**) with position along the vertical axis of the water tank. (**D**) Simulation results. Each simulated spike train was then tested to determine whether it crossed the threshold for a spatially modulated cell (see Materials and methods). Out of the 1,500 spike trains, 1,497 were classified correctly (false negative rate <0.002). The underlying data supporting all panels in

this figure can be found in a file named **S6 Fig_data.mat** (see Data Availability).
(TIF)

**S7 Fig. Occupancy correction examples. (A)** Trajectory and (**B**) an example of spiking locations (correspond to the cell shown in Fig 2I). (**C, D**) The fish's trajectory and spike positions were binned using a 5 cm × 5 cm grid to generate an occupancy map (panel C) and a spike per bin map (panel D). (**E-F**) The maps were then smoothed using a 2D Gaussian ($\sigma = 20$ cm) to obtain auxiliary maps of occupancy per bin (panel E) and the spike count per bin (panel F), respectively. A bin-by-bin division of these maps yielded the occupancy corrected heatmap as shown in Fig 2J. (**G-L**) Another example, corresponding to the cell shown in S2B Fig. The underlying data supporting all panels in this figure can be found in a file named **S7 Fig_data.mat** (see Data Availability).
(TIF)

**S1 Table. Results summary of classification of cells into distance encoding or gaussian encoding.** In correspondence with the results presented in S4K Fig.
(TIF)

**S2 Table. Number of cells classified as boundary vector cells.** Different max-p values and spatial information thresholds (see Materials and methods) were tested to show that the prevalence of boundary vector cells in the population is not strongly affected by the chosen thresholds.
(TIF)

**S1 Video. Video example of a boundary vector cell.** Left panel: A video recording shows an example of a boundary vector cell wirelessly recorded from the goldfish brain while the fish was freely exploring the experimental water tank. Spiking activity can be heard throughout the audio channel. Right panels- swimming trajectories (black curves, top panel) and spiking locations (red dots) together with the occupancy corrected rate map (bottom panel) of the cell presented in the corresponding video. This example corresponds to the cell presented in Fig 2A and 2B.
(MP4)

**S2 Video. Video example of a boundary vector cell.** Left panel: A video recording shows an example of a boundary vector cell wirelessly recorded from the goldfish brain while the fish was freely exploring the experimental water tank. Spiking activity can be heard throughout the audio channel. Right panels: Swimming trajectories (black curves, top panel) and spiking locations (red dots) together with the occupancy corrected rate map (bottom panel) of the cell presented in the corresponding video. This example corresponds to the cell presented in S2B Fig.
(MP4)

**S3 Video. Video example of a boundary vector cell.** Left panel: A video recording shows an example of a boundary vector cell wirelessly recorded from the goldfish brain while the fish was freely exploring the experimental water tank. Spiking activity can be heard throughout the audio channel. Right panels: Swimming trajectories (black curves, top panel) and spiking locations (red dots) together with the occupancy corrected rate map (bottom panel) of the cell presented in the corresponding video. This example corresponds to the cell presented in S2H Fig.
(MP4)

## Acknowledgments

We are grateful to Jacob Vecht from Deuteron Technologies and Tal Novoplansky-Tzur for helpful technical assistance.

## Author Contributions

**Conceptualization:** Lear Cohen, Ehud Vinepinsky, Opher Donchin, Ronen Segev.

**Data curation:** Lear Cohen, Ehud Vinepinsky, Ronen Segev.

**Formal analysis:** Lear Cohen, Ehud Vinepinsky, Opher Donchin, Ronen Segev.

**Funding acquisition:** Opher Donchin, Ronen Segev.

**Investigation:** Lear Cohen, Ehud Vinepinsky, Opher Donchin, Ronen Segev.

**Methodology:** Lear Cohen, Ehud Vinepinsky, Opher Donchin, Ronen Segev.

**Supervision:** Opher Donchin, Ronen Segev.

**Writing – original draft:** Lear Cohen, Ehud Vinepinsky, Ronen Segev.

**Writing – review & editing:** Lear Cohen, Ehud Vinepinsky, Opher Donchin, Ronen Segev.

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
