## [Editor Report · Decision Letter 0]

6 Jul 2022

Dear Dr Segev, 

Thank you for submitting your manuscript entitled "Axial encoding schematics of neural representations of 3D space in freely navigating goldfish" for consideration as a Research Article by PLOS Biology.

Your manuscript has now been evaluated by the PLOS Biology editorial staff [as well as by an academic editor with relevant expertise and I am writing to let you know that we would like to send your submission out for external peer review.

Once your full submission is complete, your paper will undergo a series of checks in preparation for peer review. After your manuscript has passed the checks it will be sent out for review. To provide the metadata for your submission, please Login to Editorial Manager (https://www.editorialmanager.com/pbiology) within two working days, i.e. by Jul 08 2022 11:59PM.

Kind regards,

Kris

Kris Dickson, Ph.D. (she/her)

Neurosciences Senior Editor/Section Manager

PLOS Biology

kdickson@plos.org

---

## [Decision Letter · Decision Letter 1]

17 Aug 2022

Dear Dr Segev,

Thank you for your patience while your manuscript "Axial encoding schematics of neural representations of 3D space in freely navigating goldfish" was peer-reviewed at PLOS Biology. It has now been evaluated by the PLOS Biology editors, an Academic Editor with relevant expertise, and by several independent reviewers. 

While we and the reviewers are enthusiastic about these findings in principal, Reviewers 2 and 3 in particular felt that additional experiments were needed before the conclusions could be fully substantiated. We would therefore like to invite you to revise the work to thoroughly address the reviewers' reports. You can find the detailed reviewer feedback at the end of this email.

Given the extent of revision needed, we cannot make a decision about publication until we have seen the revised manuscript and your response to the reviewers' comments. Your revised manuscript is likely to be sent for further evaluation by all or a subset of the reviewers.

**IMPORTANT - SUBMITTING YOUR REVISION**

*Re-submission Checklist*

*Published Peer Review*

*PLOS Data Policy*

*Blot and Gel Data Policy*

Sincerely,

Kris

Kris Dickson, Ph.D. (she/her)

Neurosciences Senior Editor/Section Manager

PLOS Biology

kdickson@plos.org

REVIEWS:

Reviewer's Responses to Questions

PLOS authors have the option to publish the peer review history of their article (what does this mean?). If published, this will include your full peer review and any attached files.

Reviewer #1: No

Reviewer #2: No

Reviewer #3: No

Reviewer #1: This is a very original study on navigational neurons in fish which I read with great interest and plesure. This is to say, the manuscript is well written describing up-to- date methods and ambitious, as well as demanding experiments.The authors clearly establish a new concept of 3D space representation in the telencephalon of teleosts, in this study the goldfish. Instead of the well described place cells, grid cells boundary-vector cells, and head direction cells in mammals and birds they found neurons with firing patterns that gradually increased or decreased along spatial axes distributed in all directions in 3D space. These results were obtained from freely swimming fish in a water tank by wireless tetrode recordings. The data are convincing and interprtations are modest and made safe by the statistics applied.

This a study perfectly suited for PLOS Biology and I recommend acceptance after minor revision

Specific comments:

I am irritated by the figures showing the swim paths superimposed with the action potentials and the transformation into heat maps, although it is done correctly. Sometimes it is couterintuitive to compare the firing pattern on the swim path to the heat map because the red area in the heat map do not always correpond to the number of action potentials or the other way round many spikes on the swim path lead to a blue area in the heat map. Therefor it would be helpful and clarifying in the legend of figs. 2, to clearly state the consequences (or necessity) of occupancy correction. I suggest to have both swimming path with spikes and heat map only in figure 2 and in the subsequent figures only show the heat maps. The information in the two figures is redundat anyhow.

Fig 2 D: I see no red or blue arrows or do you refere to the vertical lines ??

Fig. 3 there is some tuning in the orange curves corresponding to the orthogonal axis. Wouldn't that imply a broad gaussian activity distribution. I like the test sigmoid versus gaussian. But a better fit to sigmoid does not exclude also a a gaussian distribution. Please elaborate on this point.

Discussion

Since the transformation from a sigmoid to a gaussian function mathematically is straightforward I would invite some speculation how evolution could have changed the axis coding to place cells.

The destinction between the boundary vector cells (rats) and axis cells (fish) is difficult and at least in some examples to me seems impossible, e. g. heat maps in fig 2F, J and N

Reviewer #2: The authors present a follow-up study to a 2018 paper by the same group. The main claim of the paper is that, different from the previous paper that observed border cells, the cells in the current papers present spatial axes with different orientations. 

While the authors have performed a large number of difficult experiments, and their approach of investigating place fields in non-standard species, and particularly in 3D is certainly very interesting and worthwhile, the current presentation of the data as well as the experiments makes it difficult for the reader to follow the author's conclusions. I hope the following comments will contribute to clarify their findings.

Major comments: 

-With regards to previous experiments: a slightly larger tank, vertical instead of horizontal was used here. In how far does the visual surrounding around the tank impact the place fields? Could they just look different than in the 2018 paper due to different lab surroundings, i.e. visual cues?

More generally, how was the fish able to get oriented in the environment? What were the visual cues around the tank? Does rotating the tank where the fish is modify the tuning of the cells?.

-The main argument that these are not border cells seems to come from figure 4, where changing the border leads to changing fields. 

However, based on all the data shown, it seems that the fields are generally strongest at the borders or in the corners. A maybe more conclusive experiment could be to insert a diagonal wall (at 45 degrees) into the tank. If these are not border cells but indeed encode spatial axes, you would not expect to find a cell strongly firing along the diagonal. 

-Also, to get a better idea of the variety of the recorded cells, I would suggest to show all the recorded data in the SI (with 30 cells this could fit on 2 pages).

-I would also suggest to plot an overlap of all the place cells, to see how they cover the aquarium, that is to see the spatial covering of the tank. Showing examples that are not in the corners or at the edges would make the argument more convincing that these are not border cells (as stated around line 120). 

-Please plot the trajectories and corresponding firing rates in Fig. 2 and all other similar plots in separate figures, in many cases the red dots of the firing rate make it impossible to assess the spatial distribution. Please also show some examples in supplementary information how the densities of space and firing rate were calculated and how the calculation of the final place fields was done. Discussing how walls, corners, or center of the tan are represented, might also help following the argument that these cells encode spatial axes. 

Minor comments:

-Fig 3: C, F, I and L : please show some error band for these rate profiles.

- Can the position of the fish at maximum tuning be decoded from neural activity?

-All references are mixed up, so I was not able to follow in how far the literature is cited accurately

-The authors repeatedly mention 3D space, for example in the abstract, but the analysis is only for 2D and the third shorter axis is ignored. It would therefore be more accurate to just refer to 2D. Overall it seems that the experiments would be considerably more conclusive if a 3D tank could be used. 

-Line 105: Typo, "a geometric change [was] induced in the environment."

-Line 125: please also include these data in the suggest figure showing all recorded palce fields

-Fig 6G, missing y-axis range

-The introduction claims that the place cell is the most prominent one among the space encoding cells, I would change the wording on that, since many people might disagree with that statement. 

-Why was velocity not analyzed as in previous work? Does the firing rate depend on the direction in which the fish swims?

-How many cells were recorded at the same time and if more than one, did they ever differ in spatial selectivity?

-Supplementary Fig 3 seems to be one of the main arguments that these cells represent spatial axes. However, That fitting of a 1d sigmoid works better than a Gaussian is not convincing and likely indicates a problem with the fitting routine. There is clearly 2d information in these distributions. The 2d gaussian fit should converge to a distribution with one very wide axis, if indeed that were the best fit, so basically approaching a sigmoid, but this seems not to happen, so likely the parameters on the Gaussian were set wrongly or too limiting. 

-In figure 2 M and J, that represent diagonally tuned cells, also the trajectories of the fish don't seem to cover the corners; the trajectories themselves seem slightly diagonal, which might lead to a confound in the analysis. Why is that? Are the corners not accessible?

-More generally, is there a correlation between the spatial distribution of the trajectories and the resulting axis of the place fields? 

-Was the information along the narrow axis recorded? It would be interesting to know whether cells fire more at the borders or whether firing is equally distributed between the two walls, which would provide another strong argument for or against border cells. 

-What is the significance of beta oscillations in the authors view? It does not become clear from the discussion why this finding was presented or what it signifies in the authors view. 

Reviewer #3: There are several exciting aspects of the report, which combine to produce a very interesting dataset. These include: 1) the clarity and robustness of the spatial signal, analysed using different spatial measures. 2) The discovery of beta oscillatory modulation, particularly in the spatial cells, in this region (central telencephalon), potentially speaking to the importance of oscillatory modulation in spatial coding. To my knowledge, there has been no report of any kind reporting beta oscillations in fish, so the novelty is clear. 3) Importantly, because fish are moving in a 3-D world, the examinable set of firing responses can now be extended from rodent literature to include, like the bats, those involving the Z dimension. Indeed, exactly as those familiar with the Boundary vector cell (BVC) model would predict, the BVCs in this report include BVCs with directional tuning preferences involving the down/gravity direction or the up/anti-gravity direction. Thus a cell which fires at the bottom of the tank, but with a right-wall (or 'camera-east') bias, is a boundary vector cell which is tuned to fire at a short distance from a boundary in the down-east direction. Thus, excitingly, there is species continuity in boundary vector cells across fish and mammals. (The authors are oddly reluctant to apply what seems like the obvious BVC category label to these cells, which I mention further below.)

The paper offers a step-like increase in understanding vertebrate spatial representations, providing a vital piece of the jigsaw in terms of species-comparative knowledge, and it is surely likely to appeal to a wide readership. The authors' previous paper in Scientific Reports was more modest in terms of discovery, but has been well cited. I think this reflects the strong interest and appetite for comparative approaches to spatial cognition (insects, fish, birds, rodents, bats). The present findings certainly deliver on that front. Accordingly, all taken together, I enthusiastically recommend publication in PLOS Biology. 

There are, however, important revisions to make to the paper. In general, many details (some methods, but mostly analytical characterisations of the cell population) are missing from what might turn out to be a seminal report in an interdisciplinary journal. These details are crucial for others to replicate, contextualise, compare, and build upon the findings. Moreover, as I point out below at various points, I don't think the authors' conceptual framing of this population of spatial cells in terms of 'axial coding' is accurate or helpful, and in fact is somewhat misleading. The authors have discovered boundary vector cells in the dorsal telencephalon of goldfish, but don't seem to want to say this. My sense is that this is accidental and reflects understandable specialism in the fish rather than rodent literature. 

Title: 'Representations of 3D space' 

It is certainly true that the fish were freely moving in 3D space, but the authors do not describe representations of 3D space and this must be removed from the title and any mentions in main text. In their analysis, what they call the length dimension (20 cm), has in effect been collapsed, perfectly understandably, onto a plane.

More detailed characterisation

These details are crucial for others to replicate, contextualise, compare, and build upon the findings. 

A) Results

I would expect to see the following characterisations

Towards the early part of the results

 i) The absolute number of cells that are 'modulated by the fish's position' (p.5), alluding to how this has been defined. If the authors employ two definitions for different analyses (>0.1 bits/spike p.23 I think is the second definition), specify which one the initial "~30%" figure (p.5) refers to. It seems simplest to move the bottom paragraph on page 11 (describing the 31 passing the combined 3 criteria, the discard of 4, leaving 27 explicitly spatial cells out of 101 total) to page 5.

ii) This classification then creates two cell groups, the spatial cells, and the non-spatial cells. For these two cell groups, please provide the statistics of global firing rates (spikes/time for whole trial), on each of the three spatial measures whose combination classifies the cells (in-session stability, spatial coherence, and information rate), and finally waveform characteristics (an amplitude measure, and a spike width measure, for the largest spike on the 4 tetrode tips). For the spatial cells, characterise the maximal peak rates (shown on the top right of rate maps). 

Less crucial but informative is to divide the trials into two halves, as they have already done for the spatial stability analysis, and compare the integrated firing rate or average spatial rate for,e.g. top 30% peak rate spatial bins for first and second halves of the trial or first vs second trial. I suggest this because in the 6 examples they show of first vs second halves (fig 2), and first and second trial (fig 4), in all 6, the second half/trial shows lower firing rates. Was this a trend?

iii) Show more examples of histology with regional labelling (explain abbreviations), with atlas co-ordinates or other co-ordinate system. The region in Figure 1B is somewhat underdetermined. Can the region of the cells be described with any more precision; e.g. more posterior/anterior or ventral/dorsal portions of the central telencephalon? This is doubly important since the goldfish brain is not widely known. In the rodent literature, in describing a new cell type, it is customary to provide a summary figure showing where the spatial cells were recorded. This could be done using line drawings rather than histology photos.

Later part of Results

iv) Can they provide some analysis, even if not comprehensive, on the directionality of firing. Judging from Fig3 the firing is broadly omnidirectional, ie. taking Fig2A-D cell as an example where the 'neuron was mainly active when the fish was near the left wall of the tank' (p7) irrespective of the path direction taken by the fish near the left wall.

B) Methods

When was recording done relative to the experience of the environments? 

Geometric change manipulation (inserting the Perspex step)

a) This manipulation is surprisingly under-presented. 

b) The hypothesis testing is unclear.

Under-presentation. 1) There is no statistical analysis of the (sub)group of cells (n = ?) examined in this manipulation; 2) The authors show data for only n = 2 cells; 3) Moreover both these cells have low firing rates, so if firing is somewhat inhibited, there is not much signal to observe; 4) There is no methods description other than describing the Perspex step itself; when was recording done relative to the experience of the step? First trial, second trial? Etc How long was the trial? 

Unclear hypothesis-testing. The authors should spell out characteristics of firing they expect to see following the manipulation for each hypothesis they are testing. To me, simply put, the cells carry on behaving like boundary vector cells (BVCs), albeit with the noisy firing of low firing rate cells. The cell (figure parts I-P) which fired when it was close to the top of the tank in the baseline trial continues to do so in the manipulation trial, entirely consistently with a BVC whose firing rule is 'fire when close to a boundary in the up direction'. The firing of cell (figure parts A-F) is somewhat less clear because it is quite a low firing rate cell, and the authors autoscale its rate map to the peak in the baseline trial (ie the one with higher rate, see point above). Nevertheless, this cell which fires at the bottom of the tank in baseline continues to fire at the bottom in manipulation, but now also along the top of the Perspex step (as is clearer comparing part B to E), consistent with a BVC whose firing rule is 'fire when close to a boundary in the down direction'. There is less firing on the top of the step at the very left of the step, true, but this is obscured by autoscaling to the firing rate peak in the baseline trial, and there might also be some novelty-related inhibition of firing around the step. There is also some sign of the waveform amplitude being lower on session 2, so thresholding might be missing some spikes. The overall impression from these two cells is that the BVC model explains their firing pretty well. 

Beta Oscillation

What is the basis of the 4 db/Hz threshold? Present alternative thresholds and their statistics, or some other way to obviate picking an arbitrary threshold.

Spatial terminology

Important. The authors are not using the terms principal axis, and the related terms major and minor axes, correctly. In geometry and the disciplines using it (e.g. maths, physics, psychology of space, such as Randy Gallistel, Ken Cheng and others who suggest that to explain re-orientation behaviour in diverse animals such as fish, chicks, and rodents, animals may extract principle axes from their environment), these words have a precise meaning shared across those disciplines. Principle axes are perpendicular to each other and, importantly, pass through the centroid of the body/environment, with the major principal axis being the longest one, and minor axis (axes) being shorter than the major axis. I comment further upon the implications of this misuse of terminology below in the authors characterisation of these cells as 'axial'. It becomes clear as the report proceeds that the authors use the word 'minor axis' to mean any imaginable axis in the environment. For analytical purposes, they test 180 axes. There are not 178 potential minor axes in the standard accepted sense of that term. The authors cannot re-invent these terms' meaning any more than 'hypotenuse' or 'acute angle'. If they mean something else, ok, but then use different words.

Less important. It would help the reader to restrict the different words to apply to the same axis. At different points in the paper, the Z axis is called 'vertical', 'height' and 'depth'. Could the authors just use one verbal label consistently? It won't be obvious to readers that the Y dimension, 20 cm long in the tank, is called 'length', and the X axis is called width. Given this kind of issue, to help the reader 'fix' the authors' axis label words in mind, it would be helpful to annotate Figure 1A further, using those words, e.g. saying 'Z (height)' and 'X (width)'. The picture should mention the 'Y length' dimension, for clarity, though the Y dimension is in effect collapsed onto the plane. 

Less important. When phrases such as 'correlated to a position in the negative X direction' are used, it will be helpful to add an intuitive phrase, like 'i.e. near the left wall of the tank'. Phrases such as 'negative vertical direction' are not that obvious by themselves. 

Axial coding

I found the authors' characterisation of this population in terms of axial coding unhelpful conceptually. Further to the incorrect terminology above, I can't see what the concept is precisely, and what it adds. Its main use as far as I can tell is in opposition to the idea of a Gaussian place field type response. That the cells do not fire like place fields is very convincing but a somewhat obvious 'straw man' hypothesis to undermine. 

What is specific to the axial concept? What does the axial concept predict in further studies of these cells, ie how would they behave in such and such an environment or manipulation? 

The authors write "Activity was modulated along a preferred principal axis in the environment." "Not all cells were tuned to the major axes of space, but rather to a specified minor axis." And so on.

If the best conceptual way to characterise this cell population is that their firing is axial, one would predict at least some cells firing along the principal axes in the universally-understood sense of that phrase (see above points re 'spatial terminology'.) As mentioned above, proponents of the provocative 'geometric module' theory such as Gallistel and Cheng, whether in its harder or softer forms, have often suggested that to explain re-orientation behaviour in diverse animals such as fish, chicks, and rodents, animals may extract principle axes from their environment. The simplest prediction from such a principal axis view would be that there would be cells firing along the principal axes. However, the number of cells in this report which fire along either of the two principal axes examinable here (Z and X passing through the centroid), is n = 0. Well that is quite a disconnect. 

As the report stands, with the misleading terminology about principal, major and minor axes, casual readers might think the authors have discovered cells which in a straightforward way embody the representations of space suggested by Gallistel. They have not. This is an important issue not to be misleading about, because the rich ideas in the geometric module theory have been very influential. E.g. The initial Ken Cheng paper has >1400 citations, Gallistel's The organization of learning has nearly 5000 citations.

2) Rather, in both the main and supplementary figures, every single cell can be concisely described as firing at a particular distance and direction to boundaries in the tank, i.e. the walls, bottom and top (water/air surface), including all the cells that fire near corners (see BVCs in Lever et al, J Neuroscience, 2009, Figure 3). Thus it seems that the Occam's razor view is that this cell population is best described as a boundary vector cell population. 

3) As I have mentioned, the firing of their cell type looks like BVCs in the rodent subiculum. Another cell type in the rodent subiculum is the 'axis of travel' cell as in the (Olson et al, Nature Neuroscience, 2017) report, which in passing also finds about 21% of subiculum cells are boundary vector cells. Importantly, these rodent axis of travel cells were characterised in environments with symmetrical linear tracks. An axis of travel might fire say along half a north-south oriented track on the left of the environment, and fire along the same portion, with mirror symmetry, of the north-south oriented track on the right of the environment. Importantly, these axial travel cells lose their 'axis of travel' tuning in open field environments (ie 2D versions of the 3D open field fish tank used here.) Clearly, these goldfish cells are definitely not like the subicular axial cells, and it would be helpful not to accidentally mislead readers on this point.

References

A golden rule as a reviewer is not to mention something in the second round that was not mentioned in the first round, unless it relates to newly supplied information. Since the references were incorrectly numbered throughout the entire paper, it was not possible to assess some claims and arguments properly without very time consuming guessing, which could turn out wrong. Accordingly, once the references are corrected, there may be further, hopefully minor tweaks, to address.

---

## [Decision Letter · Decision Letter 2]

18 Jan 2023

Dear Dr Segev,

Thank you for your patience while we considered your revised manuscript "Boundary vector cells in the central telencephalon of freely navigating goldfish" for publication as a Research Article at PLOS Biology. This revised version of your manuscript has been evaluated by the PLOS Biology editors, the Academic Editor and the original reviewers.

Based on the reviews, we are likely to accept this manuscript for publication, provided you satisfactorily address the remaining points raised by Reviewer 3. We will do our best to evaluate your responses editorially with the help of our Academic Editor, aiming to avoid an additional round of peer review. 

***When revising this work, we also ask that you consider a slight title change to include some functional information in the title, to help our broad readership more easily grasp the significance of this work. Please consider something like:

Boundary vector cells in the central telencephalon of freely swimming goldfish enable navigation

OR

Boundary vector cells in the goldfish central telencephalon encode spatial information

***Please also make sure to address the data and other policy-related requests found below my signature. 

*Note that failure to fully address these points will result in delays in processing your submission.*

We expect to receive your revised manuscript within two weeks. 

*Published Peer Review History*

*Press*

Sincerely,

Kris

Kris Dickson, Ph.D., (she/her)

Neurosciences Senior Editor/Section Manager,

kdickson@plos.org,

PLOS Biology

DATA POLICY:

Note that we do not require all raw data. Rather, we ask that all individual quantitative observations that underlie the data summarized in the figures and results of your paper be made available.

1) Please provide this summary data in one of the following forms:

a) Supplementary files (e.g., excel). Please ensure that all data files are uploaded as 'Supporting Information'.

b) Deposition in a publicly available repository. Please also provide the accession code or a reviewer link so that we may view your data before publication. 

Regardless of the method selected, please ensure your files are invariably referred to (in the manuscript, figure legends, and the Description field when uploading your files) using the following format verbatim: S1 Data, S2 Data, etc. Multiple panels of a single or even several figures can be included as multiple sheets in one (e.g. excel) file that is saved using exactly the following convention: S1_Data.xlsx (using an underscore)

2) Please ensure that you provide the individual numerical values that underlie the summary data displayed in the following figure panels as they are essential for readers to assess your analysis and to reproduce it:

Fig 1D,E; Fig2-7 all panels

Supplemental Fig2 all; Fig 3C-J; Fig4-7 all panels

3) Please also ensure that figure legends in your manuscript include information on where the underlying data can be found (e.g. “The underlying data supporting Fig X, panel Y can be found in file Z.”)., and ensure your supplemental data file/s has a legend.

DATA NOT SHOWN?

- Please note that per journal policy, we do not allow the mention of "data not shown", "personal communication", "manuscript in preparation" or other references to data that is not publicly available or contained within this manuscript. Please carefully check your submission for any such statements and either remove mention of any such data or provide figures presenting the results and the data underlying the figure(s).

Reviewer remarks:

Reviewer's Responses to Questions

Do you want your identity to be public for this peer review?

Reviewer #1: No

Reviewer #2: No

Reviewer #3: No

Reviewer #1: The authors have satisfactorily responded to my comments and suggestion . I have no further requests

Reviewer #2: The authors have addressed all my concerns.

Reviewer #3: SECOND ROUND

The manuscript is greatly improved from the initial submission, and close to publication ready. 

Statistics

The authors do not provide a summary or the statistics for many of the analyses which are currently just shown as scatterplots/graphs. This is true of all of Figure 6C-H. I am not going to list all, but please attend to this. For instance, in response to this Reviewer's suggestion that, for the boundary cells, the second half of the trial may have reliably lower rates than the first half, the authors present Figure 6H. I asked "Was this a trend?". To answer should be straightforward since this data seems easily amenable to a standard paired t-test, which I imagine would suggest this trend is statistically significant. Can the authors suggest why this might be true, if so?

Directional tuning

The main issue which needs to be addressed is the swimming directional analysis, 

The results are in Lines 149-163. 

Their methods are in lines 470-484, which I quote in full here:

"Allocentric swimming direction tuning analysis. To test for the effect of allocentric swimming direction on boundary vector cells, we calculated a modulation ratio index; namely, the ratio that minimized the least square error between each cell's two tuning curves - firing rate vs. distance from the preferred boundary while swimming towards it and while swimming away from it (e.g., blue and yellow curves in Figure 4B, respectively). For this purpose, we only use the data recorded when the displacement of the fish was in the range Θ ± 45 and in the range Θ + 180 ± 45, respectively. 

To estimate the relationship between the directional and spatial information in boundary vector cells (as shown in Figure 4J), we repeated the process described in (12) to calculate the mutual information between firing rate and location and compared it to the mutual information between firing rate and the allocentric swimming direction of the fish. As in (12), we corrected the inhomogeneous sampling using exactly 48 bins (7.5 degrees each) to calculate the directional information and 48 bins on average for the spatial information."

The authors produce two results: 

1) the modulation ratio for 90 degree range centred on theta max and theta+180. That is, the authors have omitted about half of the data. The directions the authors have chosen to compare are arguably exactly the wrong ones to compare. This is due to the behavioural confound with speed and location. Although goldfish and rats are not entirely similar, there is a classic problem of behavioural dependencies, well known in rodent spatial analysis (Burgess et al, 2005, Hippocampus). Notably a head at the front of a body cannot (easily) be facing/moving away from the wall/corner, while that head is simultaneously directly abutting the wall/corner. It is also typically impossible/difficult to do this while travelling at some speed. The expectation is that you can travel at speed right into a corner, but not away from it. Such dependencies would be much less pronounced in a spheroid tank. 

2) The mutual information results show that the amount of locational information carried by the goldfish boundary cells is on average 5x the amount of their directional information. As the authors point out, this is also what Ref 12 found, ie Lever et al, 2009, where the ref 12 claim was that directionality of firing was minimal.

There is an apparent contradiction/paradox because both studies show remarkably similar mutual information results (both studies: Locational info = 5x Directional info) but this goldfish study is claiming, on the basis of the modulation ratio index, that the cells are strongly directional. Furthermore, they claim in Discussion that this is a difference from the rodent BVCs. They are not yet convincing on that point. 

Methods Differences in the studies

A) For directionality analysis, Ref 12, which like this study mainly used a rectilinear-walled box, looked at a subset of BVCs cells (60% of sample) which were also sampled in cylinders. This was because a rectilinear-walled box elicits much more behavioural dependency of the kind referred to above (head, speed, location) than cylinders. Here, the authors have the rectangular box only. 

B) The authors have not fully understood the procedure for correction of the inhomogeneous sampling in Study ref 12. They have correctly applied the same number of bins for the directional information and the locational/spatial information. That is an important equivalence. However, what ref 12 referred to regarding correcting inhomogeneous sampling was an additional analysis tool (the so-called PxD maximum likelihood algorithm in Burgess et al, 2005, Hippocampus), trying to correct for (minimise) behavioural dependencies. The authors should remove the claim they are correcting for inhomogeneous sampling with respect to behaviour. 

So what is the solution? 

The authors are free to conduct more sophisticated analysis, if they want. Supplementary Figure 5 contains a few plots which look at the speed and direction distribution, but this does not directly address the confounds. If the authors want to make the paper acceptable quickly, the minimal-work revision to make is simply to remove the modulation ratio analysis, which is at this point too simplistic. If they want to continue to argue the point strongly that they make in Figure 4, then they need to present an analysis of the firing fields which says that the swimming speeds towards the boundary are the same as those away from the boundary, within that firing field. There is also an issue if the fields are much bigger for one direction. Thus overall, it might be simplest to remove the modulation ratio, which is only based on some of the data, so not equivalent to the spatial information approach. 

Histology in Supplementary Figure 1

Please magnify, and provide some regional markers/labels on, the brain slice photo examples shown in this figure. Less important but useful: it would be simpler if the brain slices were in the same orientation as the brain maps. It is awkward for readers to do a mirror translation in their heads.

Supp Fig 5

Part D is simply a repeat of part A. This is confusing. Just alter the legend to say that Part A applies to other cells too. 

There should be a key displayed for the directions so we know what 0, -3 and +3 refer to in terms of the actual fishtank.

---

## [Decision Letter · Decision Letter 3]

21 Feb 2023

Dear Dr Segev,

Thank you for your patience while your manuscript "Boundary vector cells in the goldfish central telencephalon encode spatial information" was re-reviewed at PLOS Biology. I have now taken over the handling of your manuscript, as Kris Dickson has taken another position and is no longer with the PLOS Biology team. Your revision has now been evaluated by the PLOS Biology editors, the Academic Editor that advised on previous versions of this submission and by reviewer 3. 

Based on the review, and after discussing the file with the Academic Editor, it is clear there are some outstanding issues that persist and we thus cannot move to acceptance of your work. As you will see, the reviewer continues to raise various issues with the clarity of the figures, the reporting, the statistical analyses (including continuing to consider the claims of swimming directionality insufficiently supported) and the modulation ratio. I would urge you to be detailed and thorough in this last revision, as we will not be able to proceed until these issues are satisfactorily addressed.

In addition, Thank you for depositing all the date underlying your findings in GitHub. As you have realized, this needs to be accompanied by archiving the last version of your publicly available GitHub data to Zenodo, which will time stamp it and give it a DOI. In your case, the Zenodo file seems to be empty, so there may have been an issue. Please make sure that they you archive in Zenodo the final set of data (after this last round of revisions), and indicate in the Data Availability Statement the most updated Zenodo URL (rather than the DOI, as this does not easily resolve into the relevant file).

We expect to receive your revised manuscript within two weeks - please let us know if you would need more time.

To submit your revision, please go to https://www.editorialmanager.com/pbiology/ and log in as an Author. Click the link labelled 'Submissions Needing Revision' to find your submission record. 

Your revised submission must include the following:

*Published Peer Review History*

Please note that you may have the opportunity to make the peer review history publicly available. The record will include editor decision letters (with reviews) and your responses to reviewer comments. Please see here for more details:

*Press*

Sincerely,

Nonia

Nonia Pariente, PhD, 

Editor-in-Chief,

npariente@plos.org,

PLOS Biology

Reviewer #3 remarks:

This is an excellent set of findings, whose report still contains a few kinks, including errors. 

Figures

1) Directional scheme icon

I mentioned this before but the authors have not yet placed a circle direction icon indicating what is 0, 90, 180, 270 degrees in their scheme. Fig 1 is best for this. They are adopting zero is Camera East, and 90 is north (anti clockwise progression), like physicists/maths oriented workers. Note many assume that Zero is North and progression is clockwise (90 East), so the directional scheme needs to be clarified for readers in Figure 1 and ideally repeated in Supp fig 2 to appreciate the subsequent figures. 

2) Supp Fig 2

Looking at directions and distances more carefully in Supp Fig 2, I note that the firing rates in Supp Fig 2 are INCREASING away from the boundary, unlike in Fig 4B. The rate distance plots are surely the wrong way round. Presumably the authors need to do a mirror horizontal reflection to make them conform to Fig 4B. These should be decreasing away from the boundary as per the 4B. 

They should note somewhere in the Discussion that not ALL cells fire maximally at the boundary and decrease away, but some show peak firing at a short distance away from it. I note this is true for 5 of the 22 cells shown in sup fig 2. 

Statistics

1) Statistics and summarising fundings

The authors wrote in the previous revision

"In addition, the strength of spatial modulation could not be estimated by either the mean firing rate during the entire recording session (Figure 6F) or…"

They are now writing in the latest version that there is a correlation

"In addition, the strength of the spatial modulation was also correlated with the mean firing rate during the entire recording session (Figure 6F)."

Inspection of the figure shows that this correlation is r = -0.33. That is quite a change from saying there was no correlation. 

A key takeaway is that doing statistical tests provides a useful reality-check upon impressionistic summary. 

2) With the above caution re impressionistic summary in mind, the summary claims re swimming towards and away from the boundary are not sufficiently supported by the Figure parts 4CFLI. Note that for 3 of the 4 cells, the speed is indeed slower going away for the first (1 cell) and first and second (2 cells) closest distances, which is what matters. Equivalent speeds far away from the boundary eg say 50cm are irrelevant, since the cells do not fire far away from the boundary. It looks to me like speed could indeed be contributing to this effect. At any rate, one cannot make a claim based on 4 cells alone, when the data in those 4 cells is not that compelling, and the 4 cells are a small % of the n = 35 dataset. 

Supplementary figure 5 simply doesn't address any of these points. 

Thus the claim in lines 235-238 needs to be toned down, with more caveat phrasing, and methods mentions should make clear that the analysis has only been done on 4 cells: lines 503-6 implies a generalised analysis.

3) Please report statistics with more detail.

a) I leave this to journal discretion, but I would favour more stats description and values appearing in the main text, and not having to look at the figures all the time. It helps give a sense of what are stronger and weaker effects, and helps the reader understand exactly of what the statistical test is composed. See b and c below. 

b) It should be clear in Figure 6 that 6H is not a correlation test, but a t-test. Since a scatterplot is shown, in a context of pearson correlations, that could mislead. The values are likely correlated, but that is not the analysis done. Report t value, df. Same goes for 6G. 

c) I am unclear re precisely what on the Y axis is being correlated in 6CDEF. It looks like, and should be, the spatial tuning operationalised as the max-p value. (This helps to understand why the correlations are negative.) However, the Y axis in these 4 scatterplots is labelled as Cell hash, ie an ID number. Not sure what is happening here. Are the authors converting max-p values into a rank? If so, Pearson's correlation is not the right test. If not, relabel the Y axis appropriately. 

4) 6I is incorrectly using a linear Pearson's correlation, where it should be a circular-linear one. I am not disputing there is likely no correlation, but the authors must understand that a linear correlation test based on the Y axis having 360 at the top, and zero at the bottom is not appropriate. 350 at top is very close to 10 degrees at bottom in reality, and so forth. Best would be to show this plot in the 6B circular format. 

Modulation ratio

Using the full data plus or minus 90 not 45 is a nice improvement. I note the mean modulation ratio has dropped accordingly. 

Two points remain. 

a) Add modulation ratio values to the plots in Fig 4, so readers have intuition for this measure. 

b) Relatedly, importantly, explain if the modulation ratio is signed or unsigned. Figure 4M shows that ALL the modulation ratio values are ABOVE unity. The Methods do not make it clear whether the ratio is defined as always BLUE/TOWARDS divided by YELLOW/AWAY, or rather TOWARDS/AWAY or AWAY/TOWARDS. There is quite a difference between these possibilities. If the ratio is BLUE/TOWARDS divided by YELLOW/AWAY, then ALL the cells fire at a greater rate towards the boundary.

---

## [Editor Report · Decision Letter 4]

14 Mar 2023

Dear Dr Segev,

Thank you for the submission of your revised Research Article "Boundary vector cells in the goldfish central telencephalon encode spatial information" for publication in PLOS Biology. On behalf of my colleagues and the Academic Editor, Jozsef Csicsvari, I am pleased to say that we can in principle accept your manuscript for publication, provided you address any remaining formatting and reporting issues. These will be detailed in an email you should receive within 2-3 business days from our colleagues in the journal operations team; no action is required from you until then. Please note that we will not be able to formally accept your manuscript and schedule it for publication until you have completed any requested changes.

PRESS

Sincerely, 

Nonia

Nonia Pariente, PhD, 

Editor-in-Chief

PLOS Biology

npariente@plos.org